

# A Bayesian approach towards daily pan-Arctic sea ice freeboard estimates from combined CryoSat-2 and Sentinel-3 satellite observations

William Gregory[1], Isobel R. Lawrence[2], and Michel Tsamados[1]

[1]Centre for Polar Observation and Modelling, Earth Sciences, University College London, UK
[2]Centre for Polar Observation and Modelling, University of Leeds, UK

**Correspondence:** William Gregory (william.gregory.17@ucl.ac.uk)

**Abstract.** Observations of sea ice freeboard from satellite radar altimeters are crucial in the derivation of sea ice thickness estimates, which in turn inform on sea ice forecasts, volume budgets, and productivity rates. Current spatio-temporal resolution of radar freeboard is limited as 30 days are required in order to generate pan-Arctic coverage from CryoSat-2, or 27 days from Sentinel-3 satellites. This therefore hinders our ability to understand physical processes that drive sea ice thickness

variability on sub-monthly time scales. In this study we exploit the consistency between CryoSat-2, Sentinel-3A and Sentinel-3B radar freeboards in order to produce daily gridded pan-Arctic freeboard estimates between December 2018 and April 2019. We use the Bayesian inference approach of Gaussian Process Regression to learn functional mappings between radar freeboard observations in space and time, and to subsequently retrieve pan-Arctic freeboard, as well as uncertainty estimates. The estimated daily fields are, on average across the 2018-2019 season, equivalent to CryoSat-2 and Sentinel-3 freeboards to

within 2 mm (standard deviations < 5 cm), and cross-validation experiments show that errors in predictions are, on average, ≤ 3 mm across the same period. We also demonstrate the improved temporal variability of a pan-Arctic daily product by comparing time series of the predicted freeboards, with time series from CryoSat-2 and Sentinel-3 freeboards, across 9 sectors of the Arctic. The mean of predicted and CryoSat-2 or Sentinel-3 time series are generally consistent to within 3 mm, except for the Canadian Archipelago and Greenland, Iceland and Norwegian Seas, which show discrepancies greater than 1 cm due,

in part, to biases between CryoSat-2 and Sentinel-3 observations in these locations.

## 1 Introduction

Over the past four decades remote sensing has provided key insights into the changing state of the polar climate system. From passive microwave sensors to satellite altimeters, we have seen significant reductions in the sea ice cover for all months of the

year (Stroeve and Notz, 2018), as well as an increasing length of the summer melt season (Stroeve et al., 2014a), and a dramatic shift in the dominant ice type; moving from largely thick multi-year ice, to thinner first-year ice (Maslanik et al. 2007; Kwok



2018; Stroeve and Notz 2018). Observations of sea ice thickness have proven crucial in our ability to monitor the volumetric response of sea ice to this long term climatic change (Slater et al. 2020; Tsamados et al. 2020), as well as understanding the consequences for Arctic ecosystems, whose productivity rates are driven by the sea ice thickness distribution (Sakshaug et al.

1994; Stirling 1997; Stroeve et al. 2020 - in review). Furthermore, with continued sea ice loss we are beginning to see an upward trend in Arctic maritime activity (Wagner et al., 2020), for which timely forecasts of ice conditions are required for safe passage and cost-effective planning. Our ability to provide reliable forecasts is inherently dependent on the availability of observations which allow us to exploit sources of sea ice predictability (Guemas et al., 2016). Initialising climate models with observations of sea ice thickness for example has been shown to considerably improve seasonal sea ice forecasts compared

to those initialised with sea ice concentration (Chevallier and Salas-Mélia 2012; Doblas-Reyes et al. 2013; Day et al. 2014; Collow et al. 2015; Bushuk et al. 2017; Blockley and Peterson 2018; Schröder et al. 2019; Ono et al. 2020; Balan-Sarojini et al. 2020).

In recent decades, advancements in satellite altimetry have enabled sea ice thickness to be estimated from space. This is achieved by measuring the sea ice freeboard; that is, the height of the sea ice surface relative to the adjacent ocean, and con-

verting to thickness by assuming hydrostatic equilibrium and bulk values of the ice, ocean, and overlying snow densities (Laxon et al. 2003; Quartly et al. 2019). CryoSat-2 was the first radar altimeter launched with a specific focus on polar monitoring, and while it has been pivotal in improving our understanding of polar climate, its long repeat sub-cycle means that 30 days are required in order to generate pan-Arctic coverage (up to 88° N). The same is true for the ICESat-2 laser altimeter, launched in 2018, whose capability to estimate total freeboard, at monthly timescales, has been recently demonstrated (Kwok et al.,

2019). Other radar altimeters in operation such as Sentinel-3A and Sentinel-3B have slightly shorter repeat cycles (27 days) however only extend to 81.5° latitude. With this in mind, the maximum temporal resolution we can achieve for pan-Arctic freeboard from one satellite alone is one month. Increasing the resolution leads to a step-wise drop in spatial coverage until we arrive at 1 day of observations for which we achieve less than 20% coverage at all latitudes, for tracks averaged to a 25 km$^2$ grid spacing (Tilling et al. 2016; Lawrence et al. 2019a). This poses a significant limitation in our ability to both understand

physical processes that occur on sub-monthly timescales (Lawrence, 2019b), and in capturing the temporal variability of sea ice thickness, which could inform on sea ice forecasts.

Recent studies have found ways to improve temporal coverage through the merging of different satellite products. Ricker et al. (2017) for example, merged thickness observations from CryoSat-2 and Soil Moisture and Ocean Salinity (SMOS) satellites to produce pan-Arctic thickness fields at weekly timescales. Furthermore, Lawrence et al. (2019a) showed that radar freeboard

observations from Cryosat-2, Sentinel-3A, and Sentinel-3B can be merged to produce pan-Arctic coverage below 81.5° N every 10 days. While both of these studies are significant improvements in temporal resolution, we do not yet have a daily pan-Arctic freeboard and/or thickness product. The Pan-Arctic Ice Ocean Modeling and Assimilation System (PIOMAS) sea ice thickness model (Zhang and Rothrock, 2003) is a commonly used substitute for observations when evaluating sea ice thickness in climate models, and is available at daily resolution. PIOMAS assimilates daily sea ice concentration and sea surface

temperature fields, and despite containing no information on sea ice thickness, it has been shown to be generally consistent with in-situ and submarine observations (Schweiger et al., 2011), as well as exhibiting similar mean trends and annual cycles





to both ICESat and CryoSat-2 satellites (Schweiger et al. 2011; Laxon et al. 2013; Schröder et al. 2019). However it generally over(under)estimates thin(thick) ice regions (Schweiger et al. 2011 Stroeve et al. 2014b), giving a mis-representation of the true ice thickness distribution.

The need for a daily freeboard product based on observations then is clearly well motivated. In this study we exploit the consistency between CryoSat-2 (CS2), Sentinel-3A (S3A) and Sentinel-3B (S3B) radar freeboards (Lawrence et al., 2019a) in order to produce a gridded pan-Arctic freeboard product, hereafter referred to as CS2S3, at daily resolution between December 2018 and April 2019. Our method, Gaussian Process Regression, is a Bayesian inference technique which learns functional mappings between pairs of observation points in space and time by updating prior probabilities in the presence of new information

- Bayes' theorem. While many previous studies have reported statistical interpolation methods under a variety of names, e.g. Gaussian Process Regression (Paciorek and Schervish 2005; Rasmussen and Williams 2006), kriging (Cressie and Johannesson 2008; Kang et al. 2010; Kostopoulou 2020), objective analysis (Le Traon et al., 1997), and optimal interpolation (Ricker et al., 2017), each of these methods contain the same approach to learning functional mappings and the same key set of predictive equations. Despite this, the method is almost infinitely flexible in terms of its application and model setup. We will see how

our model differs from e.g. Ricker et al. (2017) through our choice of input-output pairs, as well as our choice of prior over functions, and approach to learning model hyperparameters (see Sect. 3).

The paper is structured as follows: Sect. 2 introduces the data sets which are used within this study, Sect. 3 outlines the method of Gaussian Process Regression and presents an example of how we achieve pan-Arctic radar freeboard estimates on any given day. In Sect. 4 we evaluate the interpolation performance through a comparison with the training inputs, and cross-validation

experiments. In Sect. 5 we provide an assessment of the improved temporal variability achieved by the use of a daily product, before finally ending with conclusions in Sect. 6.

## 2    Data

In the following section we outline the processing steps applied to CS2, S3A and S3B along-track data for generating the radar freeboard observations used as inputs to the Gaussian Process Regression model, as well as listing auxiliary data sets used.

### 2.1    Freeboard

Note that radar freeboard (the height of the radar scattering horizon above the local sea surface) is distinct from sea ice freeboard (the height of the snow/ice interface above the local sea surface). To convert radar freeboard to sea ice freeboard, a priori information on snow depth, density, and radar penetration depth are required. In this study we focus solely on radar freeboard so as not to impose new sources of uncertainty. Radar freeboard can be considered as the 'base product', from

which pan-Arctic daily estimates of sea ice freeboard and thickness can later be derived. For this study, along-track radar freeboard was derived for CS2 (SAR and SARIN), S3A and S3B in a two-stage process that is detailed in full in Lawrence et al. (2019a). First, raw Level-0 (L0) data were processed to Level-1B (L1B) waveform data using ESA's Grid Processing On Demand (GPOD) SARvatore service (Dinardo et al., 2014). At the L0 to L1B processing stage, Hamming-weighting and





zero-padding were applied, both of which have been shown to be essential for sea ice retrieval and which are not included in
the ESA standard processing at this time (Lawrence et al., 2019a). Next, L1B waveforms were processed into radar freeboard
following the methodology outlined in Lawrence et al. (2019a) (based on that of Tilling et al. (2018)) and accounting for the
Sentinel-3A and -3B (S3) retracking bias, as suggested in their conclusion. After subtracting the extra 1 cm retracking bias,
2018-19 winter-average freeboards from CS2, S3A and S3B fall within 3 mm of one another (see Lawrence 2019b). Notably,
the standard deviation on the S3A(B)-CS2 difference is comparable to the standard deviation on S3A-S3B ($\sigma$=6.0(6.0)(5.9) cm
for S3A-CS2(S3B-CS2)(S3A-S3B)). Since S3A and S3B are identical in instrumentation and configuration, differing only in
orbit, this suggests that any CS2-S3A(B) radar freeboard differences are the result of noise and the fact that the satellites sample
different sea ice floes along their different orbits, rather than due to biases relating to processing. Such consistency between
data from individual satellites permits us to combine data from all three, thus in the following methodology we propagate data
from the three satellites as a single data set.

**2.2 Auxiliary data**

In order to produce pan-Arctic estimates of radar freebaord for a given day, we need to know the maximum sea ice extent
on that day. For this we extract daily sea ice concentration fields between December 2018 and April 2019 from the National
Snow and Ice Data Center (NSIDC). We choose the NASA Team sea ice algorithm applied to passive microwave brightness
temperatures (Cavalieri et al., 1996) from the DMSP F-17 SSM/I, which are provided on a 25 x 25 km polar stereographic
grid. We then select grid cells containing sea ice as those with a concentration value $\geq 15\%$.
In Sect. 3 we distinguish between first-year ice (FYI) and multi-year ice (MYI) zones in order to perform computations related
to the model setup. For this we use the Ocean and Sea Ice Satellite Application Facility (OSI-SAF) daily ice type product
(OSI-403-b) (Eastwood, 2012), derived from DMSP/SSMIS, Metop/ASCAT and GCOM-W/AMSR-2 satellites. These data
are provided on a 10 x 10 km polar stereographic grid.
Finally, in Sect. 5 we utilise the NSIDC-defined sea ice regions (Fetterer et al., 2010) in order to perform a regional analysis of
the temporal variability of the daily CS2S3 product.

**3 Method**

In this section we outline the method of Gaussian Process Regression, and how it can be used to produce gridded pan-Arctic
radar freeboard observations on any given day. In the example presented here, we average along-track freeboard observations
from each satellite on a 25 x 25 km polar stereographic grid. We also re-grid NSIDC sea ice concentration, and OSI-SAF ice
type data to the same grid.

For a given day $t$, we have gridded freeboard observations from CS2, S3A and S3B satellites ($\mathbf{z}_1(t)$, $\mathbf{z}_2(t)$ and $\mathbf{z}_3(t)$
respectively). The number of observations from each satellite varies such that $\mathbf{z}_1(t) = \{z_1(t)_i\}_{i=1}^{n_1(t)}$, $\mathbf{z}_2(t) = \{z_2(t)_i\}_{i=1}^{n_2(t)}$,
$\mathbf{z}_3(t) = \{z_3(t)_i\}_{i=1}^{n_3(t)}$ where some, but not all, of the observations between $\mathbf{z}_1(t)$, $\mathbf{z}_2(t)$ and $\mathbf{z}_3(t)$ are co-located in space. Let





us then define $\mathbf{z}(t)$ as a $n(t)$ x 1 vector $\left(n(t) = n_1(t) + n_2(t) + n_3(t)\right)$ which is generated by concatenating the freeboard observations from each of the three satellites. Repeating this step for $\pm\,\tau$ consecutive days in a window around day $t$, we combine T $= 2\tau + 1$ number of $\mathbf{z}(t)$ vectors in order to produce a single $n$ x 1 vector $\left(n = n(t-\tau) + ... + n(t) + ... + n(t+\tau)\right)$ of all observations $\mathbf{z}$. In this case we choose $\tau = 4$, hence 9 days of observations are used in the model training. This results

in the majority of grid cells having been sampled at least once over this period, thus reducing the prediction uncertainty at any given grid cell. We can see in Fig. 1 how, in this example, the spatial coverage is improved from $\sim 23\%$ to $\sim 72\%$ by using 9 days of observations instead of 1. Our aim is then to understand the function which maps the freeboard observations to their respective space-time positions in order to make predictions at unobserved locations on day $t$. This functional mapping is commonly expressed as:

$$\mathbf{z} = f(\mathbf{x}, \mathbf{y}, \mathbf{t}) + \epsilon \quad , \quad \epsilon \sim \mathcal{N}(0, \sigma^2), \tag{1}$$

where $\epsilon$ represents independent identically distributed Gaussian noise with mean 0 and variance $\sigma^2$. The zonal and meridional grid positions of the freeboard observations are then given by the $n$ x 1 vectors $\mathbf{x}$ and $\mathbf{y}$ respectively, and finally $\mathbf{t}$ is a $n$ x 1 vector which contains the time index of each observation point $\mathbf{t} = \left(\{(t-\tau)_i\}_{i=1}^{n(t-\tau)} \,,\, ... \,,\, \{(t)_i\}_{i=1}^{n(t)} \,,\, ... \,,\, \{(t+\tau)_i\}_{i=1}^{n(t+\tau)}\right)$. For convenience we also define the collective training inputs $\boldsymbol{\Phi} = (\mathbf{x}, \mathbf{y}, \mathbf{t})$ as a $n$ x 3 matrix, such that $f(\mathbf{x}, \mathbf{y}, \mathbf{t}) \equiv f(\boldsymbol{\Phi})$.

Gaussian Process Regression (GPR) is a Bayesian inference technique which enables us to learn the function $f$ from the training set of inputs $\boldsymbol{\Phi}$ and outputs $\mathbf{z}$, and to subsequently make predictions for a new set of test inputs $\boldsymbol{\Phi}_* = (x_*, y_*, t)$. Here the test inputs correspond to the zonal and meridional grid positions of where we would like to evaluate the predictions. For now let us assume that this corresponds to one grid cell which contains sea ice on day $t$ (i.e. a grid cell within the grey mask shown in Fig. 1). In GPR we assume that $f$ is a Gaussian Process ($\mathcal{GP}$), which means that we automatically place a prior

probability over all possible functions, not just the function we wish to learn (see e.g. Rasmussen and Williams 2006). We can then assign higher probabilities to functions which we think might closely resemble $f$; which we do through a choice of mean and covariance function:

$$f(\boldsymbol{\Psi}) \sim \mathcal{GP}(m(\boldsymbol{\Psi}), k(\boldsymbol{\Psi}, \boldsymbol{\Psi}')). \tag{2}$$

Here $\boldsymbol{\Psi}$ represents arbitrary function inputs (either the training $\boldsymbol{\Phi}$ or test inputs $\boldsymbol{\Phi}_*$ in our case). For the prior mean $m(\boldsymbol{\Psi})$ we

assign a constant value which is given as the mean of CS2 FYI freeboards from the 9 days prior to the first day of training data, i.e. from $(t-\tau-9)$ to $(t-\tau-1)$. The reason that we only use FYI freeboards is that with fewer observation points, GPR has less evidence to support significantly different freeboards from $m(\boldsymbol{\Psi})$. From Fig. 1e we can see that we have fewer observation points in the (FYI) coastal margins, hence predictions here will likely remain close to $m(\boldsymbol{\Psi})$. In the MYI zone however, we have a large number of observations (more evidence) to support freeboards which may be different from $m(\boldsymbol{\Psi})$, so the choice

of $m(\boldsymbol{\Psi})$ will likely have little effect on the prediction values here. For the prior covariance $k(\boldsymbol{\Psi}, \boldsymbol{\Psi}')$, we choose the anisotropic Matérn covariance function:

$$k(\boldsymbol{\Psi}, \boldsymbol{\Psi}') = \sigma_f^2 \left(1 + \sqrt{3}d(\boldsymbol{\Psi}, \boldsymbol{\Psi}')\right) \exp\left(-\sqrt{3}d(\boldsymbol{\Psi}, \boldsymbol{\Psi}')\right), \tag{3}$$





with Euclidean distance $d(\mathbf{\Psi}, \mathbf{\Psi}')$ in space and time given by:

$$d(\mathbf{\Psi}, \mathbf{\Psi}') = \frac{\|\boldsymbol{\psi}_x - \boldsymbol{\psi}'_x\|}{\ell_x} + \frac{\|\boldsymbol{\psi}_y - \boldsymbol{\psi}'_y\|}{\ell_y} + \frac{\|\boldsymbol{\psi}_t - \boldsymbol{\psi}'_t\|}{\ell_t}.$$

In the above definition, $\sigma_f^2$, $\ell_x$, $\ell_y$, and $\ell_t$ (and also $\sigma^2$ from Eq. (1)) are known as hyperparameters, each taking a value $> 0$. $\sigma_f^2$ controls the overall variance of the function values, while $(\ell_x, \ell_y, \ell_t)$ are the space and time correlation length scales, i.e. how far does one have to move in the input space (metres or days) for the observations to become un-correlated. Let us now refer to $\boldsymbol{\theta} = (\sigma_f^2, \ell_x, \ell_y, \ell_t, \sigma^2)^{\mathrm{T}}$ as the collection of all hyperparameters. In the Bayesian approach we can select values of $\boldsymbol{\theta}$ which maximise the log marginal likelihood function:

$$\ln p(\mathbf{z}|\mathbf{\Phi}, \boldsymbol{\theta}) = -\frac{1}{2}\tilde{\mathbf{z}}^{\mathrm{T}}\mathbf{K}^{-1}\tilde{\mathbf{z}} - \frac{1}{2}\ln|\mathbf{K}| - \frac{n}{2}\ln(2\pi), \tag{4}$$

where

$$\tilde{\mathbf{z}} = \mathbf{z} - m(\mathbf{\Phi}) \quad \text{and} \quad \mathbf{K} = k(\mathbf{\Phi}, \mathbf{\Phi}') + \sigma^2 \boldsymbol{I}.$$

This procedure chooses hyperparameters in such a way as to make $p(\mathbf{z}|\mathbf{\Phi}, \boldsymbol{\theta})$ (the probability of the observations under the choice of model prior) as large as possible, across all possible sets of hyperparameters $\boldsymbol{\theta}$. It is also worth noting that Eq. (4) is

a useful tool in terms of model selection. In classical approaches one might compare different models (e.g. different choices of prior covariance function) through cross-validation, however in the Bayesian approach we can select the preferred model by evaluating Eq. (4) directly on the training set alone, without the need for a validation set. In this case we tested a variety of prior covariance functions and found the Matérn to be favourable, with the highest log marginal likelihood.

Once the hyperparameters have been found, the model is then fully determined and we can generate predictions of radar

freeboard at the test locations $\mathbf{\Phi}_*$. This amounts to updating our prior probabilities of the function values from Eq. (2), by conditioning on the freeboard observations $\mathbf{z}$ in order to generate a posterior distribution of function values. The posterior function values $\boldsymbol{f}_*$ come in the form of a predictive distribution $\boldsymbol{f}_* \sim \mathcal{N}(\bar{\boldsymbol{f}}_*, \sigma_{f_*}^2)$, where both the mean $\bar{\boldsymbol{f}}_*$ and variance $\sigma_{f_*}^2$ can be computed in closed form:

$$\begin{aligned} \bar{\boldsymbol{f}}_* &= m(\mathbf{\Phi}_*) + k(\mathbf{\Phi}, \mathbf{\Phi}_*)\mathbf{K}^{-1}\tilde{\mathbf{z}} \\ \sigma_{f_*}^2 &= k(\mathbf{\Phi}_*, \mathbf{\Phi}_*) - k(\mathbf{\Phi}, \mathbf{\Phi}_*)\mathbf{K}^{-1}k(\mathbf{\Phi}_*, \mathbf{\Phi}). \end{aligned} \tag{5}$$

Now that we have the framework in place to generate a predictive distribution of radar freeboard values for a given set of training and test locations, let us explore how we implement this in practice. Due to the need to invert a matrix of size $n$ x $n$ ($\mathbf{K}$), GPR has run time complexity $\mathcal{O}(n^3)$, which means that if we double the number of training points then we increase the run time by a factor of 8. As we need to invert $\mathbf{K}$ at every iteration step when optimising the model hyperparameters, GPR becomes increasingly computationally expensive with increasing $n$. For this reason, we take an iterative approach to the predictions:

for a particular day $t$, we generate predictions of radar freeboard at each grid cell by using only the available training data that exist within a 300 km radius (Fig. 2). We essentially repeat the process shown in Fig. 2 for every grid cell which contains sea ice, until we produce a pan-Arctic field on day $t$. While this does effectively mean that we consider observations beyond



300 km distance to be uncorrelated, it does have the advantage of both computational efficiency, and it allows us to freely incorporate spatial variation into the length scales $\ell_x$, $\ell_y$, $\ell_t$ (and hence spatial variation into the smoothness of the function

values). This seems sensible given the different scales of surface roughness that exist between FYI and MYI (Nolin et al., 2002). Furthermore, we can consider 300 km to be a reasonable distance in estimating the spatial covariance between inputs given that freeboard observations are correlated up to a distance of at least 200 km due to along-track interpolation of sea level anomalies (Tilling et al. 2018; Lawrence et al. 2018).

Fig. 3 shows the result of repeating the steps in Fig. 2 for every grid cell which contains sea ice on day $t$. The freeboard values

here corresponds to the mean of the posterior distribution $\bar{f}_*$, and the uncertainty as the square root of the variance term $\sigma_{f_*}$. Notice that the CS2S3 freeboard appears smoother than if we were to simply average the training observation points (e.g. Fig. 1e). This is because GPR estimates the function values $f$, while we have stated in Eq. (1) that the observations $\mathbf{z}$ are the function values corrupted by random Gaussian noise. We also notice how the uncertainty in our estimation of the freeboard values increases in locations where we have less training data, and is typically highest in areas where we have no data at all

(e.g. the polar hole above 88° N). In Sect. 5 we provide further discussion relating to the predictive uncertainty of the CS2S3 field, looking particularly at the Canadian Archipelago and the Greenland, Iceland and Norwegian Seas, where uncertainties are consistently higher than other regions of the Arctic.

## 4  Validation

In this section we use various metrics to assess the GPR model in terms of training and prediction. We first compare CS2S3 daily

freeboards against the training inputs in order to derive average errors across the 2018-2019 winter season, before evaluating the predictive performance of the model through cross-validation experiments. Note that hereafter, all CS2S3 fields are generated at 50 x 50 km resolution, and CS2 and S3 along-track data are averaged to the same 50 x 50 km polar stereographic grid, for computational efficiency.

### 4.1  Comparison with training inputs

Here we assess the performance of the GPR model during training; that is, an assessment of the errors between the gridded CS2S3 field, and CS2 and S3 gridded tracks, for all days between the 1st of December 2018 and the 24th of April 2019. In general, caution should be applied when evaluating training performance, as a model which fits the training data well may be fitting to the noise in the observations (over-fitting), leading to poor predictive performance. Meanwhile, a model which under-fits the training data is equally undesired (note however that over-fitting is inherently mitigated in the GPR method, see

e.g. Bishop 2006). In Eq. (1) we expressed that in our model the freeboard observations from CS2 and S3 correspond to the function values plus random Gaussian noise. Therefore by comparing our CS2S3 freeboard values from Eq. (5) to the training data, we should expect the difference to correspond to zero-mean Gaussian noise. In Fig. 4 we compute differences (CS2-CS2S3, S3A-CS2S3, S3B-CS2S3) for all days, where we can see that the difference (error) follows a normal distribution, centred approximately on zero. The average daily difference $\mu$ between CS2S3 and CS2 (or S3) is $\leq 2$ mm, with standard





deviation on the difference $\sigma < 5$ cm. Furthermore, we can see that the Root Mean Square Error (RMSE) between CS2S3 and the training inputs is equivalent to the standard deviation of the difference, which can only occur when the average bias is approximately zero. Notably, each of the standard deviations is approximately equal to the $\sim 6$ cm uncertainty on 50 km grid-averaged freeboard measurements, determined from a comparison of S3A and S3B data during tandem phase of operation (see supplementary Fig. S1). A breakdown of average errors for each month, including RMSE, are given in Table 1.

## 4.2 Cross-validation

We have seen in the previous section that CS2S3 freeboards closely resemble CS2 and S3 observations (the training data), however this does not give an indication as to how well the model performs in predicting at unobserved locations. An independent set of radar freeboard observations would allow CS2S3 freeboards to be validated at locations unobserved by CS2 or S3 on any given day, however in the absence of these we rely on alternative metrics to evaluate the predictions. Commonly, data from the airborne mission Operation IceBridge are used to validate sea ice freeboard and/or thickness observations, however as only 6 days were collected in a small area around north of Greenland in 2019, there are insufficient data points here to realistically draw any conclusions about the CS2S3 freeboards on larger temporal and spatial scales. $K$-fold cross-validation is a useful tool when validation data are limited, however models must be run $K$ number of times (ideally where $K = n$), incurring significant computational expense when $n$ is large (as it is in our case). We opt for a pragmatic solution here: we test the predictive performance of the model by removing different combinations of each of the S3 satellites from the training data, re-generating the predictions with the remaining subset, and subsequently evaluating predictions on the withheld set. For example, in the first instance we generate daily pan-Arctic predictions across the 2018-2019 period, except that we withhold both S3A and S3B from the training set. We then evaluate predictions from each day against observations from S3A and S3B. In the next instance, we repeat the same process except that we withhold only S3A from the training set (whilst retaining S3B), and use S3A as the validation set. In the final scenario, we withhold S3B from the training set and use S3B as the validation set. Fig. 5 compares predictions for each of these scenarios, across all days in the 2018-2019 winter period. As we would expect, the spread of error in the predictions is largest in the case where only CS2 is used to train the model, with $\sigma = 0.061$ m (it is worth noting again that this is approximately consistent with the error between S3A and S3B in tandem phase, as discussed in Sect. 4.1 and Supplementary 1). The spread of error between the predictions and observations then decreases slightly with the incorporation of either S3A or S3B ($\sigma = 0.059$ m). The mean error across all validation tests is consistently $\leq 3$ mm, showing that the model is able to make reliable predictions at unobserved locations. We can therefore be confident that with the inclusion of all 3 satellites, the predictions at unobserved locations are at least as good as $\leq 3$ mm error, if not better. A breakdown of equivalent statistics is presented for each month in Table 2, although we do not include RMSE here as it is again consistent with the standard deviation in each case.





## 5 Assessment of temporal variability

Finally, to showcase the capabilities of the CS2S3 product, we turn to an assessment of temporal variability. Within 9 sectors of the Arctic we calculate the mean CS2S3 freeboard for each day of the 2018-2019 winter season to produce a time series of radar freeboard evolution for each sector. We also plot the 31-day running mean freeboard from CS2, S3A and S3B to demonstrate the increase in variability when moving from a monthly to a daily product. The 9 Arctic sectors are taken from NSIDC (Fetterer et al., 2010) and include Baffin and Hudson Bays, Greenland, Iceland and Norwegian (GIN), Kara, Laptev,

East Siberian (ESS), Chukchi, Beaufort Seas, the Canadian Archipelago (CAA), and the Central Arctic.

In Fig. 6 we can see how the day-to-day variability is increased with the CS2S3 product, compared to the CS2 and S3 31-day running means. Generally the mean of the CS2S3 time series lies within 3 mm of CS2 and S3, however large discrepancies exist in the GIN Seas (up to 1.1 cm) and the CAA (up to 1.3 cm). This is perhaps not surprising given the larger uncertainty on

radar freeboard in shallow-shelf seas and at coastal margins, relating to higher uncertainty in interpolated sea level anomalies (Lawrence et al., 2019a). Indeed, we can see that the difference between the 31-day running means (CS2-S3A(B)) in the CAA is also large, at 1.1(1.9) cm. The GIN region includes the area east of Greenland, one of the most dynamic regions which carries MYI exported out of the Fram Strait southward. Being such a dynamic region, we expect that the difference in spatial sampling of the three satellites may drive discrepancies between the 31-day running means. Note that both the GIN Seas and the CAA

also coincide with where we see some of the largest uncertainty in the CS2S3 daily field (e.g. Fig. 3b); which is, in part, a reflection of the discrepancies between CS2 and S3 freeboards in these locations, but also due to the limited amount of data available in these regions across the 9-day training period. We can see from Fig. 7 that the number of training points used to inform on predictions in the coastal margins (including the CAA and GIN Seas) is, on average, less than $\sim 200$ points, whereas in the central regions (Central Arctic, Beaufort, Chukchi, ESS, Laptev, and Kara Seas), the number of training points is typically

$> 500$. A natural question is then whether the variability we see in the time series in Fig. 6 represents real physical signal, or is just noise related to observational uncertainty. To address this question we also show a 'benchmark' time series in Fig. 6. For this, we first compute a 'static' background field, which is given as the average of all CS2 and S3 gridded tracks between the 1$^{st}$ of December 2018 and the 24$^{th}$ of April 2019, and then re-generate predictions for each day, as per our GPR model, except that the training data now sample from the background field along the CS2 and S3 track locations of the respective days used

in the model training. We should therefore expect the benchmark predictions for each day to be approximately equivalent in areas where there are a significant amount of training data (see Fig 7), and to see variability in areas where there are less data, which can be explained by tracks sampling different locations of the background field on different days. This is typically what we see in Fig 6, where there is almost zero day-to-day variability of the benchmark time series in regions such as the Beaufort, Chukchi, ESS and Laptev Seas, but slight variability in the coastal margins.

## 6 Conclusions


In this study we presented a methodology for deriving daily gridded pan-Arctic radar freeboard estimates through Gaussian Process Regression; a Bayesian inference technique. We showed an example of how our method uses 9 days of gridded freeboard





observations from CryoSat-2 (CS2), Sentinel-3A (S3A) and Sentinel-3B (S3B) satellites in order to model spatio-temporal co-variances between observation points, and make pan-Arctic predictions of radar freeboard, with uncertainty estimates, on any

given day. We refer to this product as CS2S3. We also introduced a Bayesian approach for estimating the (hyper)parameters which define the covariance function; one which maximises the probability of the observations under the choice of model prior. An evaluation of the model performance was then carried out for both training and predictions. For the training points, we compared our CS2S3 freeboards to gridded CS2 and S3 freeboards at co-located points for all days across the 2018-2019 winter season, where we found the differences to follow a normal distribution, with mean errors $\leq 2$ mm, and standard deviations

$< 5$ cm. To evaluate the predictive performance of the model we performed an approach based on cross-validation where we withheld different combinations of S3 freeboards from the training set of observations, re-generated the daily predictions and used the withheld data to validate the predictions. We found the general prediction error to also be normally distributed, with mean errors consistently $\leq 3$ mm, and standard deviations $\sim 6$ cm. Finally, we presented the improved temporal variability of a daily pan-Arctic freeboard product by comparing time series of CS2S3 freeboards, with 31-day running means from CS2

and S3 observations, in 9 different Arctic sectors. We showed that the mean of the CS2S3 time series were generally within 3 mm of the 31-day running mean time series from CS2 and S3, except for the Candian Archipelago and the Greenland Iceland and Norwegian Seas, where prediction uncertainty is large due to significant discrepancies between CS2 and S3 freeboards. We then presented a benchmark test to conclude that the variability we see in our time series analysis is related to real physical signal rather than noise related to observational uncertainty. The improved temporal variability from a daily product is a hope-

ful prospect for improving the understanding of physical processes that drive radar freeboard and/or thickness variability on sub-monthly timescales. Of course, an investigation into the drivers of the temporal variability in the CS2S3 field would be an additional way to validate the product, although this goes beyond the scope of our study. We conclude here that the Gaussian Process Regression method is an extremely robust tool for modelling a wide range of statistical problems, from interpolation of geo-spatial data sets, as presented here and in other works (Le Traon et al. 1997; Ricker et al. 2017), to time series fore-

casting (Rasmussen and Williams 2006; Sun et al. 2014; Gregory et al. 2020). The Gaussian assumption holds well in many environmental applications, and the fact that the covariance structure can take any form, so long as $\mathbf{K}$ is a symmetric positive semi-definite matrix, means that the model can be tailored very specifically to the problem at hand.

*Author contributions.*    WG developed the GPR algorithm and ran the processing for generating the daily CS2S3 fields. IRL conducted all of the pre-processing of CryoSat-2 and Sentinel-3 data files, as well as performing the analysis of the Sentinel-3 tandem phase comparisons.

IRL also wrote section 2.1 of this manuscript. WG wrote the remaining sections. MT suggested the application of the GPR algorithm for this framework of combining satellite altimetry products, and also provided technical support and input for writing this manuscript. Both co-authors contributed to all sections of the manuscript.

*Competing interests.*    The authors declare that they have no conflict of interest.





*Acknowledgements.* WG acknowledges support from the UK Natural Environment Research Council (NERC) (Grant NE/L002485/1). IRL

is supported by the UK NERC Centre for Polar Observation and Modelling. MT acknowledges support from the NERC "PRE-MELT" (Grant

NE/T000546/1) and "MOSAiC" (Grant NE/S002510/1) projects and from ESA's "CryoSat+ Antarctica Ocean" (ESA AO/1-9156/17/I-BG)

and "EXPRO+ Snow" (ESA AO/1-10061/19/I-EF) projects.



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

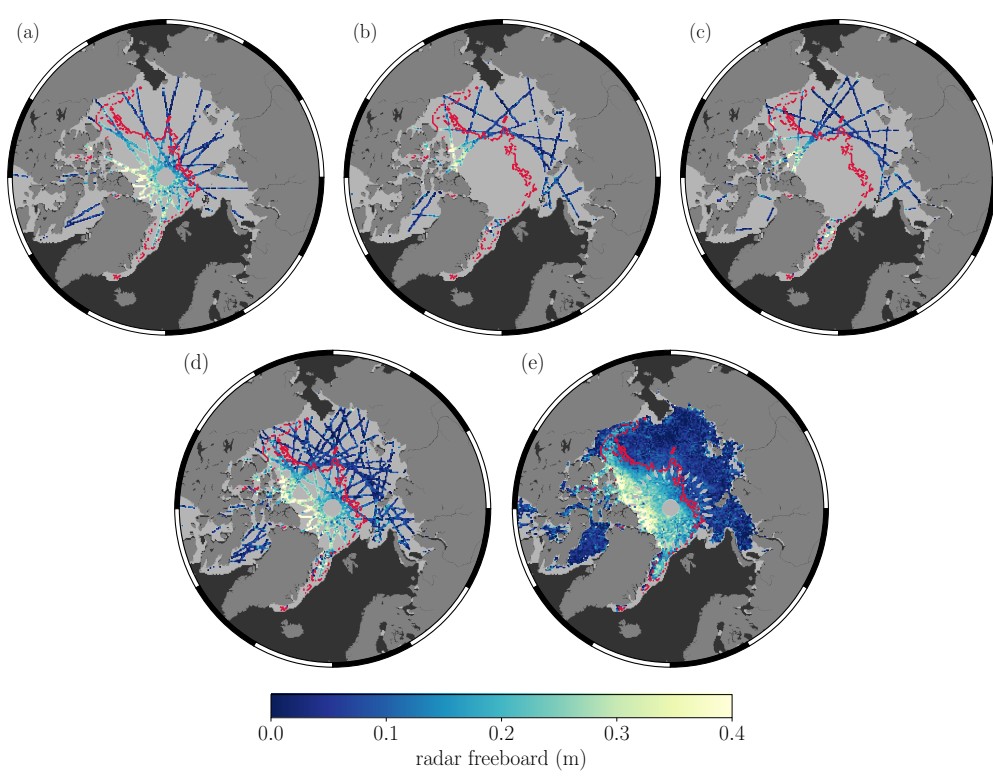

**Figure 1.** Gridded tracks from CS2 (a), S3A (b) and S3B (c) on a 25 x 25 km polar stereographic grid, respectively covering approximately 11%, 7% and 7% of the total NSIDC NASA Team sea ice extent on day $t$ (grey mask). By combining the three satellites (d), this coverage is increased to approximately 23%. Combining $t \pm 4$ days of gridded tracks (e), the coverage is increased further to approximately 72% of the total sea ice extent. The red ice type contour (from OSI-SAF) shows the boundary between thick MYI and thin FYI.

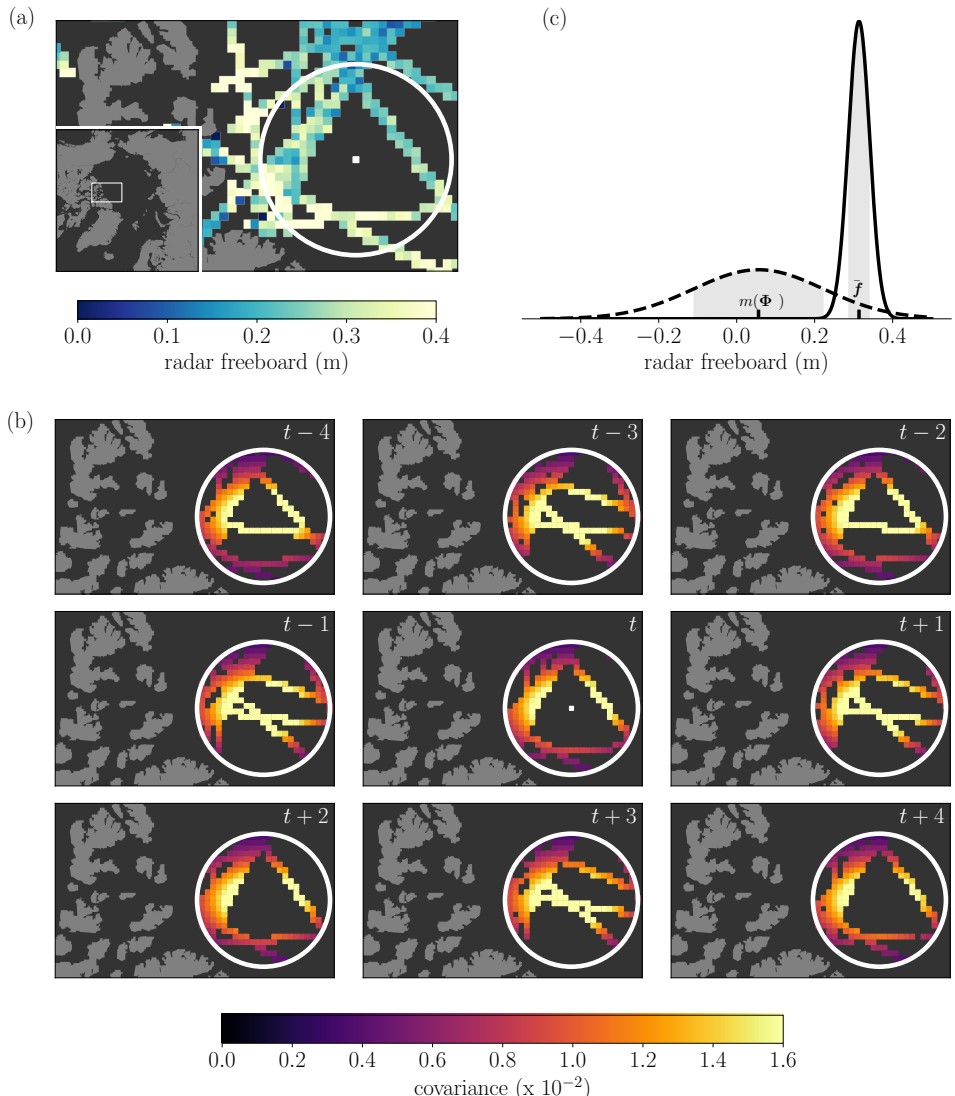

**Figure 2.** Estimating freeboard for 1 grid cell. (a) In this case we want to estimate a posterior distribution of freeboard values at the location of the white pixel on day $t$. The white circle corresponds to a distance of 300 km from the pixel, and the gridded tracks are the CS2 and S3 observations on day $t$. (b) The prior covariance $k(\mathbf{\Phi}_*, \mathbf{\Phi})$; which is the covariance between the test input $\mathbf{\Phi}_*$ (the white pixel), and all the training inputs $\mathbf{\Phi}$ that lie within a 300 km radius, for each of the 9 days of training data. (c) Probability density functions showing the prior distribution of function values (dashed), with mean $m(\mathbf{\Phi}_*) = 0.056$ m and one standard deviation $\sqrt{k(\mathbf{\Phi}_*, \mathbf{\Phi}_*)} = 0.167$ m, as well as the posterior predictive distribution (solid) showing the estimated freeboard value at the location of the white pixel on day $t$ as $\bar{f}_* = 0.314$ m, with one standard deviation of $\sigma_{f_*} = 0.028$ m.

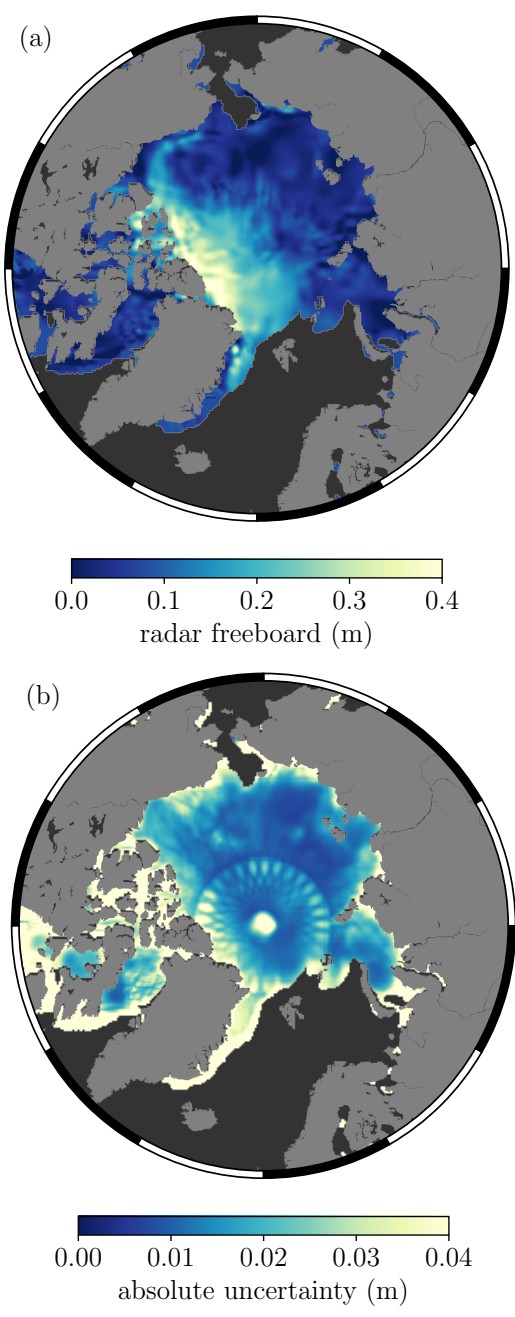

**Figure 3.** (a) Gridded CS2S3 radar freeboard for one day, from Gaussian Process Regression. (b) The absolute uncertainty (one standard deviation), corresponding to the square root of the predictive variance from Eq. (5).

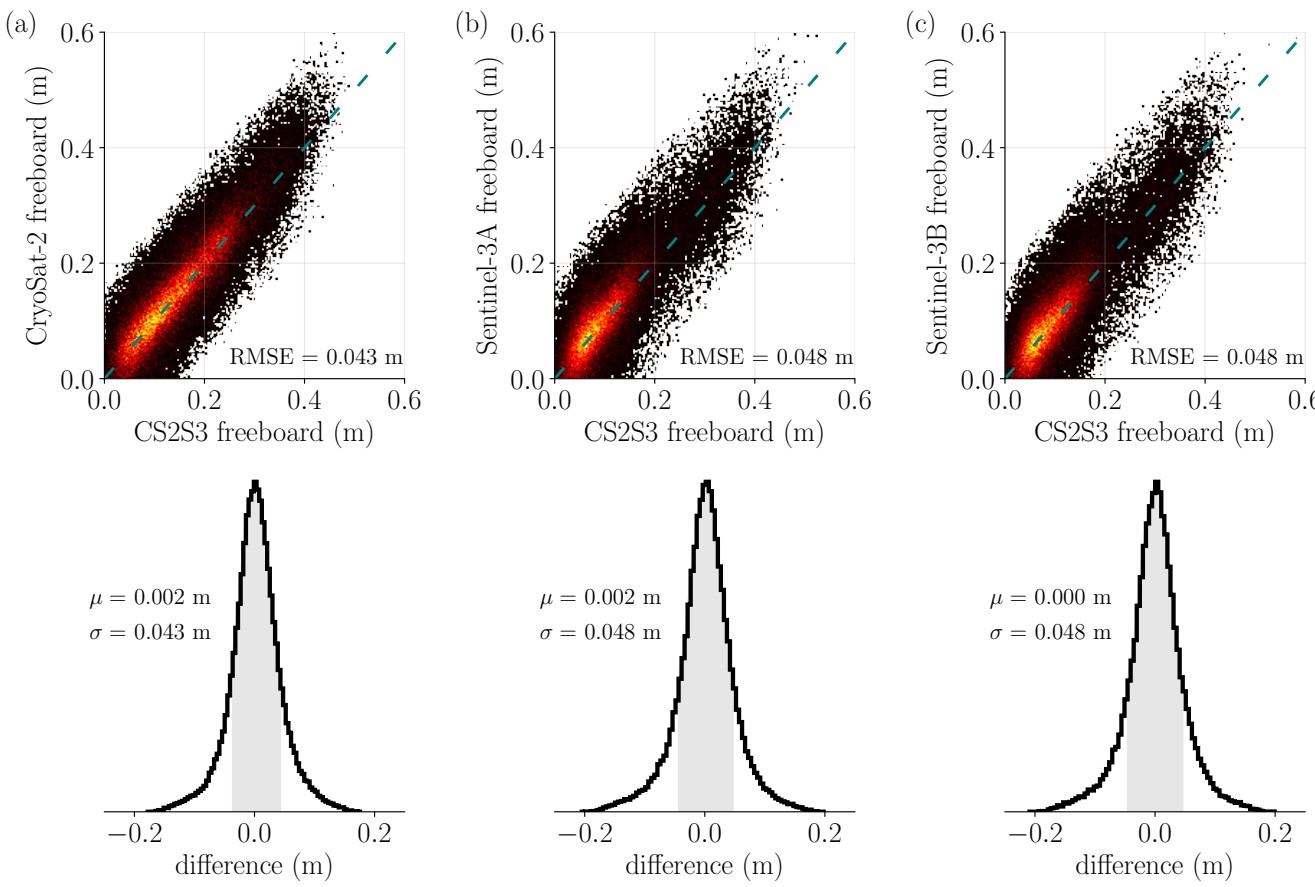

**Figure 4.** Comparison of daily gridded CS2S3 freeboard against daily CS2 (a), S3A (b), and S3B (c) gridded freeboard tracks, for all days between the 1$^{st}$ of December 2018 and the 24$^{th}$ of April 2019. In each case, the freeboard values are directly compared in the scatter-density plot (dashed line shows $y = x$). The distribution of the error is given by the histograms below each scatter plot (1$\sigma$ either side of the mean $\mu$ is shaded in grey). Only values within $\pm 3\sigma$ of the mean are shown for all plots.

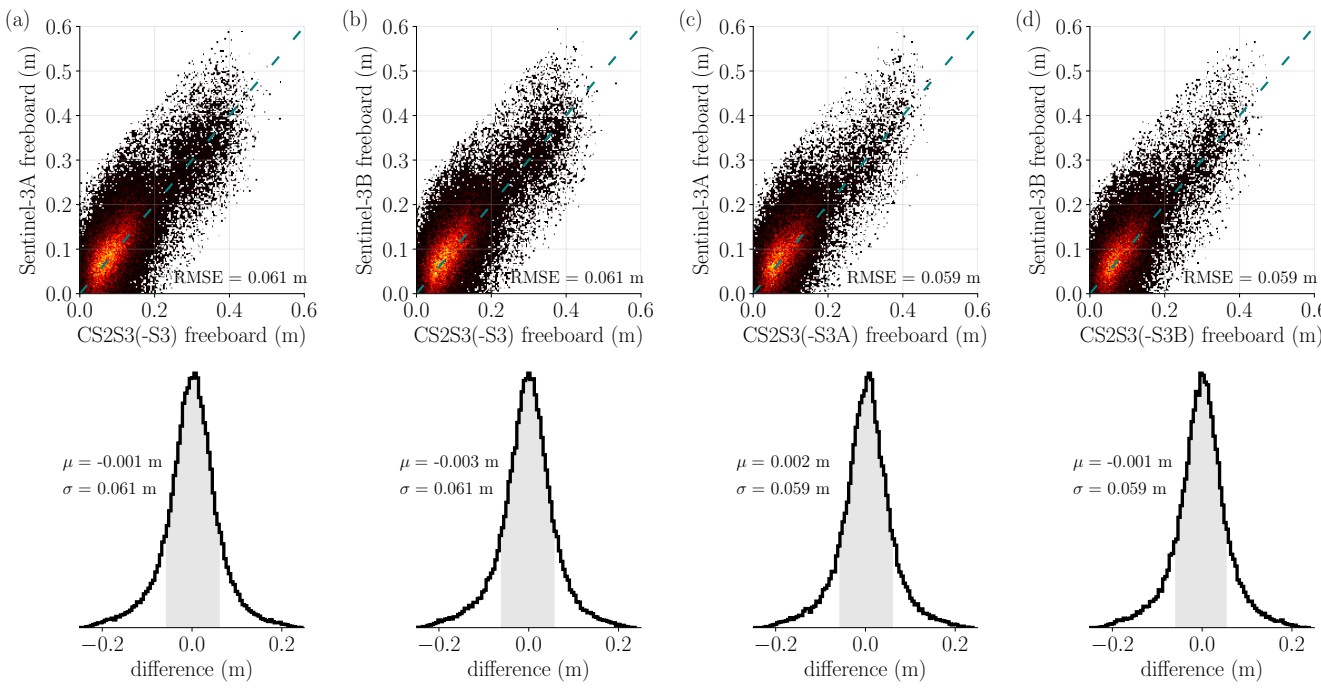

**Figure 5.** Scatter-density plots and error distributions of the GPR predictions for all days between the 1$^{st}$ of December 2018 and the 24$^{th}$ of April 2019, with different combinations of S3 satellites removed from the training data. (a) Model trained using only CS2 observations, validated against S3A. (b) Model trained using only CS2 observations, validated against S3B. (c) Model trained using only CS2 and S3B observations, validated against S3A. (d) Model trained using only CS2 and S3A obsevations, validated against S3B. $1\sigma$ either side of the mean $\mu$ is shaded in grey for each histogram. Only values within $\pm3\sigma$ of the mean are shown for all plots.



**Figure 6.** Comparison of CS2S3 time series and 31-day running means of CS2 and S3 across 9 Arctic sectors. Sectors are defined based on Fetterer et al. (2010), and are available from NSIDC. Values presented for each graph correspond to the mean (in metres) of each time series. All means are computed across the period 11$^{th}$ of December 2018 to 12$^{th}$ of April 2019 (the period where all four time series are available). Note that S3 time series are not included in the Central Arctic as much of this region overlaps with the S3 polar hole $> 81.5°$ N. Grey lines are the benchmark time series to test the efficacy of the model.

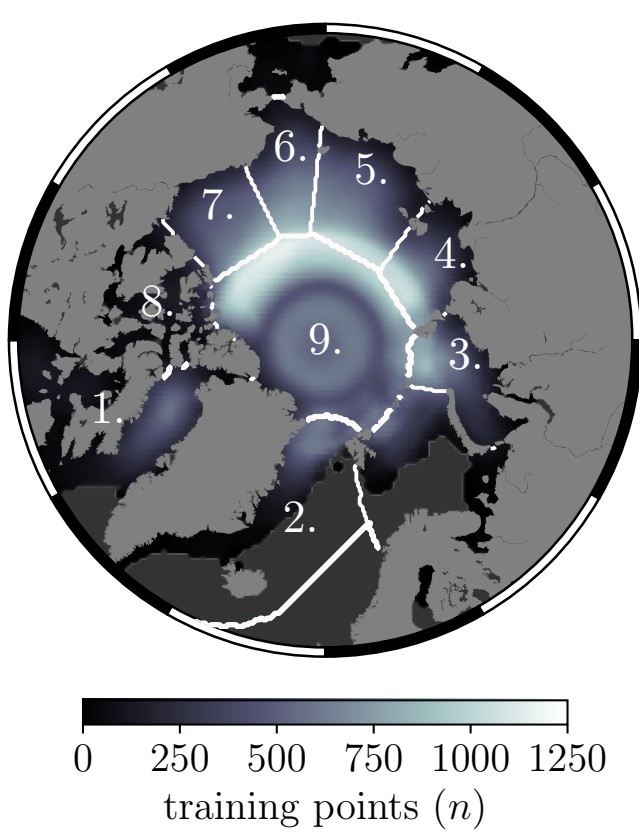

**Figure 7.** The number of training points used to predict radar freeboard at each grid cell at 50 km$^2$ resolution, shown as an average across all days between the 1$^{st}$ of December 2018 and the 24$^{th}$ of April 2019. The contour lines mark the boundaries between the 9 NSIDC regions: 1. Baffin and Hudson Bays, 2. GIN Seas, 3. Kara Sea, 4. Laptev Sea, 5. East Siberian Sea, 6. Chukchi Sea, 7. Beaufort Sea, 8. CAA, 9. Central Arctic.



**Table 1.** Mean ($\mu$), standard deviation ($\sigma$), and RMSE of the daily difference between gridded CS2, S3A and S3B tracks and co-located CS2S3 points, for all days between the 1[st] of December 2018 and the 24[th] of April 2019. Also computed for all days in each respective month.

| Date | $\mu$ (m) CS2-CS2S3 | $\sigma$ (m) CS2-CS2S3 | $\mu$ (m) S3A-CS2S3 | $\sigma$ (m) S3A-CS2S3 | $\mu$ (m) S3B-CS2S3 | $\sigma$ (m) S3B-CS2S3 | RMSE (m) CS2-CS2S3 | RMSE (m) S3A-CS2S3 | RMSE (m) S3B-CS2S3 |
|---|---|---|---|---|---|---|---|---|---|
| 201812 | 0.003 | 0.043 | 0.001 | 0.048 | 0.000 | 0.048 | 0.043 | 0.048 | 0.048 |
| 201901 | 0.002 | 0.041 | 0.002 | 0.047 | 0.000 | 0.047 | 0.041 | 0.047 | 0.047 |
| 201902 | 0.002 | 0.042 | 0.001 | 0.046 | 0.000 | 0.046 | 0.042 | 0.046 | 0.046 |
| 201903 | 0.002 | 0.042 | 0.001 | 0.048 | 0.000 | 0.048 | 0.042 | 0.048 | 0.048 |
| 201904 | 0.002 | 0.045 | 0.002 | 0.050 | 0.000 | 0.051 | 0.045 | 0.050 | 0.051 |
| all months | 0.002 | 0.043 | 0.002 | 0.048 | 0.000 | 0.048 | 0.043 | 0.048 | 0.048 |

**Table 2.** Mean ($\mu$) and standard deviation ($\sigma$) of the daily difference between gridded S3A and S3B tracks and co-located CS2S3 points, with different combinations of S3 observations removed from the training data.

| Date | $\mu$ (m) S3A-CS2S3(-S3) | $\sigma$ (m) S3A-CS2S3(-S3) | $\mu$ (m) S3B-CS2S3(-S3) | $\sigma$ (m) S3B-CS2S3(-S3) | $\mu$ (m) S3A-CS2S3(-S3A) | $\sigma$ (m) S3A-CS2S3(-S3A) | $\mu$ (m) S3B-CS2S3(-S3B) | $\sigma$ (m) S3B-CS2S3(-S3B) |
|---|---|---|---|---|---|---|---|---|
| 201812 | -0.001 | 0.061 | -0.002 | 0.060 | 0.002 | 0.059 | -0.001 | 0.059 |
| 201901 | 0.000 | 0.060 | -0.003 | 0.059 | 0.002 | 0.058 | -0.002 | 0.058 |
| 201902 | -0.001 | 0.059 | -0.002 | 0.058 | 0.001 | 0.057 | -0.001 | 0.057 |
| 201903 | -0.001 | 0.062 | -0.003 | 0.063 | 0.001 | 0.060 | -0.002 | 0.060 |
| 201904 | 0.000 | 0.065 | -0.003 | 0.065 | 0.003 | 0.061 | -0.002 | 0.062 |
| all months | -0.001 | 0.061 | -0.003 | 0.061 | 0.002 | 0.059 | -0.001 | 0.059 |