# Peer review of "A Bayesian approach towards daily pan-Arctic sea ice freeboard estimates from combined CryoSat-2 and Sentinel-3 satellite observations"

_The Cryosphere, 2020_

## Referee Comment (RC1)

**Review of "A Bayesian approach towards daily pan-Arctic sea ice freeboard estimates from combined CryoSat-2 and Sentinel-3 satellite observations"**
By Gregory et al.

**General comments**

In "A Bayesian approach towards daily pan-Arctic sea ice freeboard estimates from combined CryoSat-2 and Sentinel-3 satellite observations" the authors investigate the use of Bayesian inference to produce daily gridded pan-Arctic radar freeboard estimates. Gaussian Process Regression (GPR) is used to model spatio-temporal covariances between observations made by three ESA's satellite altimetry missions (CryoSat-2, Sentinel-3A, and Sentinel-3B) and to make pan-Arctic predictions of radar freeboard, with uncertainty estimates, on a given day.

This is a novel, interesting and relevant investigation as it attempts, for the first time, to estimate freeboard with a daily temporal resolution based solely on satellite altimetry data. The improved temporal resolution of pan-Arctic freeboard could contribute to our ability to understand physical processes driving sea ice thickness variability on sub-monthly time scales.

The study is generally well structured and the manuscript is clear and pleasant to read. I recommend this paper for publication, however, there are some points that should be addressed by the authors first.

**Specific comments**

**Data**

- Why did you choose data between December 2018 and April 2019? By selecting e.g. the following season (2019/20), you could have included in the analysis the months of October and November and make your results representative for an entire Arctic winter season.

- L85-90: Hamming-weighting and zero-padding are both applied to CS2 L0 processing (https://wiki.services.eoportal.org/tiki-download_wiki_attachment.php?attId=4431&page=Cryosat%20Documents&download=y). Please amend this statement and, if CS2 L0 data are processed using GPOD, please state the differences with the official Baseline-D version provided by ESA.

**Method**

- How do you treat observations from different satellites in the same grid cell acquired on the same day (i.e. co-located in both space and time)? Do you include these as separated inputs or do you feed them as a single averaged estimate to the GPR algorithm? This should be clarified in the manuscript.

- As there is no general "Discussion" section, I add this comment here. Bayesian inference allows to estimate the optimal covariance function hyperparameters based entirely on data as the parameters maximising the log marginal likelihood function. Do you think that the tool you developed could be useful in investigating the spatial and temporal correlation length scales of freeboard measurements? Please add a short paragraph discussing this possibility.

**Validation**

- How do you think a different grid resolution would affect your results in Section 4, e.g., by using a 25x25 km grid instead? Also, please repeat in the conclusions that the validation presented in Section 4 is based on a 50x50 km grid.

- The results in Table 1 show a slight but systematically lower freeboard mean difference between CS2S3 and S3B compared with CS2S3-CS2 and CS2S3-S3A. While rounding might play a role in this comparison, do you have any idea why CS2S3 tends to best fit S3B data for every month of your analysis?

- I understand the authors' choice of the cross-validation method, however, I think that both section 4.2 and the conclusions should clearly state that the given estimates of prediction error are based only on validation data from regions below 81.5°N and with a sea ice concentration larger than 75%, since these correspond to areas where the absolute uncertainty is usually the lowest (exception made for the Canadian Archipelago and the Fram Strait, as the authors nicely point out in Section 5). Regions above 81.5°N and with ice concentration between 15% and 75% (including the marginal ice zone) are systematically left out of the cross-validation since:
  - only S3 data are used as a validation
  - according to Lawrence et al. (2019a), diffuse waveforms within grid cells with ice concentration lower than 75% are discarded, which means that no freeboard estimates are available from any of the satellites on a given day where ice concentration falls below 75%.

- I would have expected a more significant difference in performance when training the model with CS2 data only, given the lower spatio-temporal coverage when compared with a combined CS2/S3A/S3B training data set. According to your results, a GPR based on CS2 observations alone is able to predict radar freeboard at unobserved locations pretty well (with a 3-4% RMSE increase, from 5.9 to 6.1 cm, when compared to the multi-satellite solution). Do you think this is related to the relatively coarse (50x50 km) grid chosen for your cross-validation? I suggest to add a paragraph in your discussion elaborating on this matter and on the actual advantage of including S3 data in your model training compared with using only CS2 data. In the light of these results, it would also be interesting to discuss the possibility of using data from the three satellites while reducing the number of days used for model training.

**Assessment of temporal variability**

- This is a nice section highlighting daily variations of regional freeboard estimates and larger discrepancies between CS2S3 predictions and satellite data for sectors like the GIN and the CAA. I suggest to add a couple of statements about the 'Baffin & Hudson' sector. While the average CS2 and S3 freeboard over the entire period agree within 5 mm, they show differences of ~1 cm in December 2018 and March 2019. What do you think might be the reason for this more significant, with respect to other sectors, difference?

- I would rephrase line 253 to reflect that the comparison of mean freeboard estimates over the entire observational period performed in this section is mainly a confirmation of your cross-validation results—the average value of a time series alone does not say a lot about temporal variability. I suggest something like: "... Generally, the mean of the CS2S3 time series lies within 3 mm of CS2 and S3, in line with the results of the cross-validation presented in section 4.2. However, ...".

**Technical corrections**

- L43: according to Lawrence et al. (2019a), the CS2 daily Arctic coverage is lower than 20% up to 82-83°N, not at all latitudes. Also, Tilling et al. (2016) shows Arctic coverage down to a minimum of two days, not one. Please amend this sentence to reflect the content of the cited publications
- L66/378: the DOI provided for Rasmussen and Williams (2006), a book, points to an article by Matthias Seeger with same title. Please correct the reference
- L104: if you want to be consistent with the platform/sensor notation used for the OSI SAF product, this line should perhaps read: "... from the Nimbus-7/SMMR, DMSP/SSM/I, and DMSP/SSMIS, which are ..." → (see https://nsidc.org/data/nsidc-0051 for reference)
- L108: you probably mean OSI-403-c? The 403-b product has been superseded and did not include AMSR-2 data
- L138: "For now..." → "For now, ..."
- L190: "corresponds" → "correspond"
- L291: add comma after "Greenland" → "... and the Greenland, Iceland and Norwegian Seas, ..."
- L301: I suggest not use "K" in the final statement → "... and the fact that the covariance structure can take any form, so long as the covariance matrix is symmetric, positive, and semi-definite, means ..."
- Figure 1: please state which day the sea ice concentration, type (FYI/MYI boundary) and radar freeboard refer to in the example
- Figure 3: please add the grid resolution (25x25 km) and the day which the radar freeboard estimates and uncertainty correspond to
- Figure 6: if the benchmark time series is not explained in the caption, please add a reference to the section 5
- Figure 7: please write the name of the sectors in full and provide the abbreviations, when used in the text and/or in Figure 6, in parentheses

---

## Referee Comment (RC2)

**Summary**

This paper presents a novel approach for estimating daily sea ice freeboard across the Arctic, using the Bayesian inference approach of Gaussian Process Regression (GPR). Benefits of such an approach include 1.) fewer days of data required to estimate Arctic-wide sea ice freeboard and 2.) improved temporal variability of a daily Arctic-wide freeboard product. It's great to see data from CryoSat-2, Sentinel-3A and Sentinel-3B being utilized together in this way. It was also refreshing to read a paper that has a good understanding of the relevant literature and other's methodology, so thank you to the authors for that. I do have a few concerns that should be addressed before publication, and these are summarized in my General and Specific Comments below.

**General Comments**

My one major concern with the paper was the limited assessment of improvements in temporal variability of daily sea ice freeboard (presented in Section 5). The authors show in In Fig. 6 we can see how the day-to-day variability is increased with the CS2S3 product, compared to the CryoSat-2 and Sentinel-3 31-day running means. Then, lines 265-266 state that "A natural question is then whether the variability we see in the time series in Fig. 6 represents real physical signal, or is just noise related to observational uncertainty". I'd suggest that this isn't just a natural question, but really the main question, and one I had throughout the paper until this point. It's really the crux of "why bother doing this work"? While I appreciate the development of innovative methods for improving sea ice products, the reader still needs to be sold on its benefits.

Although Figure 6 is very interesting, I'd like to see a more quantitative assessment of the temporal variability from GPR and monthly running means, compared with the benchmark (especially in regions where there are less training data). How much of an improvement in "true" temporal variability does GPR provide? The authors have clearly done most if not all of the relevant work, so please expand. Then, add mention of this in the abstract to strengthen the importance of the study.

**Specific Comments**

P1 L19: "reductions in the sea ice cover" is too general a statement. Specify what each instrument measures and over what time frame. We haven't seen reduction in thickness from altimetry for four decades, or in summer. We also haven't seen reduction in extent from altimetry. So, please be more specific here to avoid confusion.

P2 L23: Are AGU talks suitable references (I'm not sure on TC's stance on this)? If so, please provide a link to the publicly available version of the talk.

P2 L31: I'd consider Allard et al. (2018) to be a key paper that's missing here

P2 L35: Snow depth is also assumed

P2 L43: I believe Tilling et al. (2016) was 2 days

P2 L55: Change "containing" to "assimilating", for clarity

P3 L60: This first sentence doesn't add anything. I suggest using at this point to highlight the benefits of the GPR method vs. a monthly moving average that is very simple the produce. It's not just the need for a daily freeboard product based on observations that is well motivated, but a daily freeboard product that more accurately represents temporal variability. This new approach can (in theory) provide both.

P3 L75: For me, Section 5 is currently insufficient at providing "an assessment of the improved temporal variability achieved by the use of a daily product". See my General Comments above.

P4 L99: Introduce the "CS2S3" acronym here

P4 L120: On average, what percentage are co-located?

**References**

Allard, R. A., Farrell, S. L., Hebert, D. A., Johnston, W. F., Li, L., Kurtz, N. T., et al. (2018). Utilizing CryoSat-

2 sea ice thickness to initialize a coupled ice-ocean modeling system. *Advances in Space*

*Research*, *62*(6), 1265–1280.

Tilling, R. L., Ridout, A., & Shepherd, A. (2016). Near Real Time Arctic sea ice thickness and volume from

CryoSat-2. *The Cryosphere*, *10*, 2003–2016.

---

## Author Comment (AC2)

**Review of "A Bayesian approach towards daily pan-Arctic sea ice freeboard estimates from combined CryoSat-2 and Sentinel-3 satellite observations"** By Gregory et al.

**General comments**

In "A Bayesian approach towards daily pan-Arctic sea ice freeboard estimates from combined CryoSat-2 and Sentinel-3 satellite observations" the authors investigate the use of Bayesian inference to produce daily gridded pan-Arctic radar freeboard estimates. Gaussian Process Regression (GPR) is used to model spatio-temporal covariances between observations made by three ESA's satellite altimetry missions (CryoSat-2, Sentinel-3A, and Sentinel-3B) and to make pan-Arctic predictions of radar freeboard, with uncertainty estimates, on a given day.

This is a novel, interesting and relevant investigation as it attempts, for the first time, to estimate freeboard with a daily temporal resolution based solely on satellite altimetry data. The improved temporal resolution of pan-Arctic freeboard could contribute to our ability to understand physical processes driving sea ice thickness variability on sub-monthly time scales.

The study is generally well structured and the manuscript is clear and pleasant to read. I recommend this paper for publication, however, there are some points that should be addressed by the authors first.

Thank you for your kind words, and for taking the time to review our work! It is very much appreciated. Please see our comments below, which we hope address your concerns.

**Specific comments**

**Data**

- Why did you choose data between December 2018 and April 2019? By selecting e.g. the following season (2019/20), you could have included in the analysis the months of October and November and make your results representative for an entire Arctic winter season. The choice to perform our analysis for the 2018-2019 season was initially to compare with the final Operation Icebridge campaign in April 2019, however as we state in the manuscript, it was difficult to draw any conclusions based on so few data points. Note that we do plan to run this algorithm for future seasons and make the data publicly available in the near future.
- L85-90: Hamming-weighting and zero-padding are both applied to CS2 L0 processing • (https://wiki.services.eoportal.org/tikidownload wiki attachment.php?attId=4431&page=Cryosat%20Documents&download=y). Please amend this statement and, if CS2 L0 data are processed using GPOD, please state the differences with the official Baseline-D version provided by ESA. We have not compared the GPOD-derived CryoSat-2 radar freeboard with the ESA L2 baseline D product, however in Lawrence et al. (2019) they applied the same L1B -> L2 processing to GPOD L1B and ESA L1B (baseline C) data and found a radar freeboard difference of ~6mm attributed to the fact that the GPOD L1B data does not contain the stack standard deviation (SSD) parameter which is used for filtering lead and floe waveforms in the ESA L1B -> L2 processing chain. As the authors remark in their paper, it was more important to ensure consistency between CS2 and S3 than consistency between our CS2 radar freeboard and the ESA L2 freeboard product. However we agree that it is important to comment on this and we will note this difference in the revised manuscript and say that our combined product is preliminary and awaits the availability of ESA L2 Sentinel-3 freeboard which is processed in a consistent way to CS2.

**Method**

 How do you treat observations from different satellites in the same grid cell acquired on the same day (i.e. co-located in both space and time)? Do you include these as separated inputs or do you feed them as a single averaged estimate to the GPR algorithm? This should be clarified in the manuscript.

Observations which are co-located in space and time are treated as separate inputs. The GPR framework assumes that these observations are independent random samples drawn from the same distribution (i.e., the same posterior function we are trying to learn), yet have independent noise contents. We will make this clearer in the revised manuscript.

 As there is no general "Discussion" section, I add this comment here. Bayesian inference allows to estimate the optimal covariance function hyperparameters based entirely on data as the parameters maximising the log marginal likelihood function. Do you think that the tool you developed could be useful in investigating the spatial and temporal correlation length scales of freeboard measurements? Please add a short paragraph discussing this possibility.

Indeed, for each grid cell we do retain the learned hyperparameters which maximise the log marginal likelihood function. This therefore allows us to construct spatial maps of each hyperparameter (including space and time correlation length scales - see Figure 1 below). We agree that this information might be useful to potential end-users of this product and so will include a discussion on this in the revised manuscript.

Figure 1 showing the zonal (X), meridional (Y) and temporal (T) freeboard correlation length scales which maximise the log marginal likelihood function, for each grid cell when generating predictions for the 1st of December 2018.

**Validation**

How do you think a different grid resolution would affect your results in Section 4, e.g., by using a 25x25 km grid instead? Also, please repeat in the conclusions that the validation presented in Section 4 is based on a 50x50 km grid.
In Figure 2 below, we show an example of the training error distribution for one day (1st of December 2018), where interpolations were run at 25, 50 and 100 km spatial resolution. Here we can see that increasing/decreasing the resolution does not result in a systematic

increase or decrease in the average error. We do however see that the spread in error is larger for finer resolutions, which is perhaps expected as the coarser resolution data will have averaged out much of the noise content within the data. On this note, we think that it is worth including some sensitivity tests as supplementary material for this manuscript, including tests where we vary the spatial grid resolution of the input data, and also vary the number of days in the training – see more below.

Figure 2 showing the error distributions between training data (CS2 and S3) and CS2S3 interpolated freeboards for the 1st of December 2018. Each distribution shows the error for interpolations performed at different spatial resolutions, with (a) 25x25 km, (b) 50x50 km, (c) 100x100 km.

• The results in Table 1 show a slight but systematically lower freeboard mean difference between CS2S3 and S3B compared with CS2S3-CS2 and CS2S3-S3A. While rounding might play a role in this comparison, do you have any idea why CS2S3 tends to best fit S3B data for every month of your analysis?

Having since gone back and checked our calculations we have noticed a small error in the derivation of the mean and standard deviation statistics presented in Figures 4 and 5, and Tables 1 and 2. The revised statistics are given below for Tables 1 and 2:

| Table 1    | μ       | σ       | μ       | σ       | μ       | σ       | RMSE    | RMSE    | RMSE  |
|------------|---------|---------|---------|---------|---------|---------|---------|---------|-------|
| Date       | CS2-    | CS2-    | S3A-    | S3A-    | S3B-    | S3B-    | CS2-    | S3A-    | S3B-  |
| Ducc       | CS3S3   | CS3S3 |
| 201812     | 0.001   | 0.051   | 0.000   | 0.057   | -0.001  | 0.057   | 0.051   | 0.057   | 0.057 |
| 201901     | 0.001   | 0.049   | 0.001   | 0.056   | -0.002  | 0.055   | 0.049   | 0.056   | 0.055 |
| 201902     | 0.000   | 0.050   | 0.000   | 0.055   | -0.001  | 0.055   | 0.050   | 0.055   | 0.055 |
| 201903     | 0.001   | 0.050   | 0.000   | 0.056   | -0.001  | 0.057   | 0.050   | 0.056   | 0.057 |
| 201904     | 0.001   | 0.053   | 0.000   | 0.061   | -0.001  | 0.061   | 0.053   | 0.061   | 0.061 |
| all months | 0.001   | 0.051   | 0.000   | 0.057   | -0.001  | 0.057   | 0.051   | 0.057   | 0.057 |
| Table 2    | μ       | σ       | μ       | σ       | μ       | σ       | μ       | σ       |       |
| Date       | S3A-    | S3A-    | S3B-    | S3B-    | S3A-    | S3A-    | S3B-    | S3B-    |       |
| Dutt       | CS3S3(- |       |
|            | S3)     | S3)     | S3)     | S3)     | S3A)    | S3A)    | S3B)    | S3B)    |       |
| 201812     | -0.002  | 0.073   | -0.004  | 0.072   | 0.001   | 0.072   | -0.002  | 0.072   |       |
| 201901     | -0.001  | 0.071   | -0.004  | 0.071   | 0.002   | 0.070   | -0.003  | 0.070   |       |
| 201902     | -0.002  | 0.072   | -0.003  | 0.071   | 0.000   | 0.071   | -0.002  | 0.070   |       |
| 201903     | -0.003  | 0.074   | -0.005  | 0.075   | 0.000   | 0.072   | -0.004  | 0.0073  |       |
| 201904     | -0.002  | 0.079   | -0.005  | 0.076   | 0.001   | 0.076   | -0.003  | 0.076   |       |
| all months | -0.002  | 0.074   | -0.004  | 0.073   | 0.001   | 0.072   | -0.003  | 0.072   |       |

We now notice that CS2S3 freeboards are generally higher than S3B (given by the negative bias for both training and cross-validation comparisons, across all months). The model now appears to fit S3A better than S3B. Rounding does indeed play a role in these statistics, for example, if we increase the number of significant figures for the 'all months' cases  $\mu_{CS2-CS2S3}$  and  $\mu_{S3A-CS2S3}$  in Table 1, we see that  $\mu_{CS2-CS2S3} = 0.00078$  m and  $\mu_{S3A-CS2S3} = 0.00024$  m. Hence these round to 1 mm and 0 mm respectively. To address the question as to whether the difference in mean between any of the error distributions is significant (e.g., between  $\mu_{CS2-CS2S3}$  and  $\mu_{S3A-CS2S3}$ ), we can use a statistical Z-test. This can be computed through the following equation:

$$Z = \frac{\mu_{\text{CS2-CS2S3}} - \mu_{\text{S3A-CS2S3}}}{\sqrt{\frac{\sigma_{\text{CS2-CS2S3}}^2}{n_1} + \frac{\sigma_{\text{S3A-CS2S3}}^2}{n_2}}}$$

where  $n_1$  and  $n_2$  are the number of samples which make up the CS2-CS2S3 and S3A-CS2S3 error distributions respectively. The Z-test allows us to determine whether, based on the available samples from CS2-CS2S3 and S3A-CS2S3, the true means of the two error distributions are likely to be the same (i.e., the true zero-mean Gaussian noise distribution). Note that the Z-test assumes that samples are independent random variables – which is what assume the noise to be. In the example above we find that Z is equal to 2.38, which is equivalent to >99% significance. We therefore do not have evidence to reject the null hypothesis here, and can conclude that the two true means are highly likely to be the same.

- I understand the authors' choice of the cross-validation method, however, I think that both section 4.2 and the conclusions should clearly state that the given estimates of prediction error are based only on validation data from regions below 81.5.N and with a sea ice concentration larger than 75%, since these correspond to areas where the absolute uncertainty is usually the lowest (exception made for the Canadian Archipelago and the Fram Strait, as the authors nicely point out in Section 5). Regions above 81.5.N and with ice concentration between 15% and 75% (including the marginal ice zone) are systematically left out of the cross-validation since:
  - o only S3 data are used as a validation
  - according to Lawrence et al. (2019a), diffuse waveforms within grid cells with ice concentration lower than 75% are discarded, which means that no freeboard estimates are available from any of the satellites on a given day where ice concentration falls below 75%.

Thank you for raising this crucial point. We will make sure the manuscript reflects this in the revised version.

I would have expected a more significant difference in performance when training the model with CS2 data only, given the lower spatio-temporal coverage when compared with a combined CS2/S3A/S3B training data set. According to your results, a GPR based on CS2 observations alone is able to predict radar freeboard at unobserved locations pretty well (with a 3-4% RMSE increase, from 5.9 to 6.1 cm, when compared to the multi-satellite solution). Do you think this is related to the relatively coarse (50x50 km) grid chosen for your cross-validation? I suggest to add a paragraph in your discussion elaborating on this matter and on the actual advantage of including S3 data in your model training compared with using only CS2 data. In the light of these results, it would also be interesting to discuss the possibility of using data from the three satellites while reducing the number of days used for model training.

With regards to the benefits of including Sentinel-3 data during the model training, we do see clear improvements in the derived freeboard estimates (see Figure 3 below). In particular, we notice how without S3 data, features such as the 'monkey tail' in the Beaufort sea are less well defined, and in some cases interpolation artefacts are present (particularly the CS2S3(-S3) case). Furthermore, we also importantly see reduced uncertainty in freeboard by the inclusion of all satellites in the training (see Figure 4 below).

With regards to reducing the number of days for model training, we generated sensitivity tests where we ran interpolations usin

---

## Author Comment (AC3)

**Summary**

This paper presents a novel approach for estimating daily sea ice freeboard across the Arctic, using the Bayesian inference approach of Gaussian Process Regression (GPR). Benefits of such an approach include 1.) fewer days of data required to estimate Arctic-wide sea ice freeboard and 2.) improved temporal variability of a daily Arctic-wide freeboard product. It's great to see data from CryoSat-2, Sentinel-3A and Sentinel-3B being utilized together in this way. It was also refreshing to read a paper that has a good understanding of the relevant literature and other's methodology, so thank you to the authors for that. I do have a few concerns that should be addressed before publication, and these are summarized in my General and Specific Comments below.

Thank you very much for your enthusiastic feedback, and for taking the time to review our work! Please see our comments below, which we hope address your concerns.

**General Comments**

My one major concern with the paper was the limited assessment of improvements in temporal variability of daily sea ice freeboard (presented in Section 5). The authors show in In Fig. 6 we can see how the day-to-day variability is increased with the CS2S3 product, compared to the CryoSat-2 and Sentinel-3 31-day running means. Then, lines 265-266 state that "A natural question is then whether the variability we see in the time series in Fig. 6 represents real physical signal, or is just noise related to observational uncertainty". I'd suggest that this isn't just a natural question, but really the main question, and one I had throughout the paper until this point. It's really the crux of "why bother doing this work"? While I appreciate the development of innovative methods for improving sea ice products, the reader still needs to be sold on its benefits. Although Figure 6 is very interesting, I'd like to see a more quantitative assessment of the temporal variability from GPR and monthly running means, compared with the benchmark (especially in regions where there are less training data). How much of an improvement in "true" temporal variability does GPR provide? The authors have clearly done most if not all of the relevant work, so please expand. Then, add mention of this in the abstract to strengthen the importance of the study.

Determining whether our interpolation algorithm captures 'real' daily freeboard variability is an important feature of this work, so thank you for helping us to strengthen this section. Although we use the benchmark to ensure that the signal does not originate from spatial sampling, it is difficult to identify what exactly is causing the daily temporal variability. One thing we know to alter the radar freeboard is snowfall, therefore (building on the work of Lawrence 2019), we have compared our daily freeboard timeseries to snowfall from ERA5 to see whether any correlation emerges – see Figure 1 below. Here we break down the analysis into first-year-ice (FYI) and multi-year-ice (MYI), since the roughness and snow depth are typically different for the two ice types (Tilling et al., 2018) and therefore variability of radar freeboard with snowfall may also show different signals. We can see here that daily CS2S3 freeboard anomalies are positively correlated with ERA5 snowfall data, with an average Pearson correlation of 0.4 for FYI and MYI zones. While it goes beyond the scope of this study to investigate the specific drivers of this correlation (see Lawrence 2019 for a discussion on why radar freeboard correlates with snowfall at 9-day timescales), we believe that this relationship is encouraging, as it suggests that the CS2S3 data are able to capture freeboard variability at synoptic time scales. We will include these results in the revised manuscript, and update the abstract accordingly.

References
Lawrence, I. R.: Multi-satellite synergies for polar ocean altimetry, Ph.D. thesis, UCL (University College London), 2019.

Tilling, R.L., Ridout, A. and Shepherd, A., 2018. Estimating Arctic sea ice thickness and volume using CryoSat-2 radar altimeter data. *Advances in Space Research*, *62*(6), pp.1203-1225.

[Figure]

Figure 1 showing daily CS2S3 radar freeboard anomalies, with daily ERA5 snowfall data for FYI and MYI zones (spatial extent of ice zones are given by the accompanying map, where colours are consistent). Pearson correlations (r) are shown for each plot

**Specific Comments**

We will amend all of the specific comments below in the revised manuscript.

P1 L19: "reductions in the sea ice cover" is too general a statement. Specify what each instrument measures and over what time frame. We haven't seen reduction in thickness from altimetry for four decades, or in summer. We also haven't seen reduction in extent from altimetry. So, please be more specific here to avoid confusion.

P2 L23: Are AGU talks suitable references (I'm not sure on TC's stance on this)? If so, please provide a link to the publicly available version of the talk.

P2 L31: I'd consider Allard et al. (2018) to be a key paper that's missing here

P2 L35: Snow depth is also assumed

P2 L43: I believe Tilling et al. (2016) was 2 days

P2 L55: Change "containing" to "assimilating", for clarity

P3 L60: This first sentence doesn't add anything. I suggest using at this point to highlight the benefits of the GPR method vs. a monthly moving average that is very simple the produce. It's not just the need for a daily freeboard product based on observations that is well motivated, but a daily freeboard product that more accurately represents temporal variability. This new approach can (in theory) provide both.

P3 L75: For me, Section 5 is currently insufficient at providing "an assessment of the improved temporal variability achieved by the use of a daily product". See my General Comments above.

P4 L99: Introduce the "CS2S3" acronym here

P4 L120: On average, what percentage are co-located?

**References**

Allard, R. A., Farrell, S. L., Hebert, D. A., Johnston, W. F., Li, L., Kurtz, N. T., et al. (2018). Utilizing CryoSat-2 sea ice thickness to initialize a coupled ice-ocean modeling system. Advances in Space Research, 62(6), 1265–1280.

Tilling, R. L., Ridout, A., & Shepherd, A. (2016). Near Real Time Arctic sea ice thickness and volume from CryoSat-2. The Cryosphere, 10, 2003–2016.

We would like to again thank the reviewer for their invested time in reviewing this work.

---

## Author Response (AR1)

**Reviewer Comments 1**

**Review of "A Bayesian approach towards daily pan-Arctic sea ice freeboard estimates from combined CryoSat-2 and Sentinel-3 satellite observations"**
By Gregory et al.

**General comments**

In "A Bayesian approach towards daily pan-Arctic sea ice freeboard estimates from combined CryoSat-2 and Sentinel-3 satellite observations" the authors investigate the use of Bayesian inference to produce daily gridded pan-Arctic radar freeboard estimates. Gaussian Process Regression (GPR) is used to model spatio-temporal covariances between observations made by three ESA's satellite altimetry missions (CryoSat-2, Sentinel-3A, and Sentinel-3B) and to make pan-Arctic predictions of radar freeboard, with uncertainty estimates, on a given day.

This is a novel, interesting and relevant investigation as it attempts, for the first time, to estimate freeboard with a daily temporal resolution based solely on satellite altimetry data. The improved temporal resolution of pan-Arctic freeboard could contribute to our ability to understand physical processes driving sea ice thickness variability on sub-monthly time scales.

The study is generally well structured and the manuscript is clear and pleasant to read. I recommend this paper for publication, however, there are some points that should be addressed by the authors first.

Thank you for your kind words, and for taking the time to review our work! It is very much appreciated. Please see our comments below, which we hope address your concerns.

**Specific comments**

**Data**

- Why did you choose data between December 2018 and April 2019? By selecting e.g. the following season (2019/20), you could have included in the analysis the months of October and November and make your results representative for an entire Arctic winter season.
  The choice to perform our analysis for the 2018-2019 season was initially to compare with the final Operation Icebridge campaign in April 2019, however as we state in the manuscript, it was difficult to draw any conclusions based on so few data points. Note that we do plan to run this algorithm for future seasons and make the data publicly available in the near future.

- L85-90: Hamming-weighting and zero-padding are both applied to CS2 L0 processing (https://wiki.services.eoportal.org/tiki-download_wiki_attachment.php?attId=4431&page=Cryosat%20Documents&download=y). Please amend this statement and, if CS2 L0 data are processed using GPOD, please state the differences with the official Baseline-D version provided by ESA.
  Thank you for pointing this out. We have revised the statement regarding Hamming-weighting and zero-padding (please see L90-L93 of the revised manuscript). We have also included a brief paragraph detailing some of the work from Lawrence et al 2019a, which compared L1B → L2 processing, between GPOD and ESA (L102-L108).

**Method**

- How do you treat observations from different satellites in the same grid cell acquired on the same day (i.e. co-located in both space and time)? Do you include these as separated inputs or do you feed them as a single averaged estimate to the GPR algorithm? This should be clarified in the manuscript.

Observations which are co-located in space and time are treated as separate inputs. The GPR framework assumes that these observations are independent random samples drawn from the same distribution (i.e., the same posterior function we are trying to learn), yet have independent noise contents. We have since made this clearer in the revised manuscript (L138-L141) – see also RC2 comment about co-located observations.

- As there is no general "Discussion" section, I add this comment here. Bayesian inference allows to estimate the optimal covariance function hyperparameters based entirely on data as the parameters maximising the log marginal likelihood function. Do you think that the tool you developed could be useful in investigating the spatial and temporal correlation length scales of freeboard measurements? Please add a short paragraph discussing this possibility.

  Indeed, for each grid cell we do retain the learned hyperparameters which maximise the log marginal likelihood function. This therefore allows us to construct spatial maps of each hyperparameter (including space and time correlation length scales). We have included a brief discussion on this in the revised manuscript (L214-L231), along with an accompanying Figure (Fig. 4) of each of the correlation length scales.

**Validation**

- How do you think a different grid resolution would affect your results in Section 4, e.g., by using a 25x25 km grid instead? Also, please repeat in the conclusions that the validation presented in Section 4 is based on a 50x50 km grid.

  We would expect that varying the grid resolution does not lead to systematic differences in error. As a way to illustrate this, we have included a supplementary figure (Fig. S3) showing the training error for 1 day, for interpolations generated at 25x25, 50x50, and 100x100 km grid resolution. We notice here that the mean error is not systematically higher/lower for higher/lower grid resolutions. A reference to this figure can be found in the main text on L249. We have also updated the conclusions to state that the results from Section 4 are generated at 50x50 km resolution (see L344).

- The results in Table 1 show a slight but systematically lower freeboard mean difference between CS2S3 and S3B compared with CS2S3-CS2 and CS2S3-S3A. While rounding might play a role in this comparison, do you have any idea why CS2S3 tends to best fit S3B data for every month of your analysis?

  Having since gone back and checked our calculations we have noticed a small error in the derivation of the mean and standard deviation statistics presented in Figures 4 and 5, and Tables 1 and 2. The revised statistics are given below for Tables 1 and 2:

| Table 1 Date | $\mu$ CS2-CS3S3 | $\sigma$ CS2-CS3S3 | $\mu$ S3A-CS3S3 | $\sigma$ S3A-CS3S3 | $\mu$ S3B-CS3S3 | $\sigma$ S3B-CS3S3 | RMSE CS2-CS3S3 | RMSE S3A-CS3S3 | RMSE S3B-CS3S3 |
|---|---|---|---|---|---|---|---|---|---|
| 201812 | 0.001 | 0.051 | 0.000 | 0.057 | -0.001 | 0.057 | 0.051 | 0.057 | 0.057 |
| 201901 | 0.001 | 0.049 | 0.001 | 0.056 | -0.002 | 0.055 | 0.049 | 0.056 | 0.055 |
| 201902 | 0.000 | 0.050 | 0.000 | 0.055 | -0.001 | 0.055 | 0.050 | 0.055 | 0.055 |
| 201903 | 0.001 | 0.050 | 0.000 | 0.056 | -0.001 | 0.057 | 0.050 | 0.056 | 0.057 |
| 201904 | 0.001 | 0.053 | 0.000 | 0.061 | -0.001 | 0.061 | 0.053 | 0.061 | 0.061 |
| all months | 0.001 | 0.051 | 0.000 | 0.057 | -0.001 | 0.057 | 0.051 | 0.057 | 0.057 |

| Table 2 Date | $\mu$ S3A-CS3S3(-S3) | $\sigma$ S3A-CS3S3(-S3) | $\mu$ S3B-CS3S3(-S3) | $\sigma$ S3B-CS3S3(-S3) | $\mu$ S3A-CS3S3(-S3A) | $\sigma$ S3A-CS3S3(-S3A) | $\mu$ S3B-CS3S3(-S3B) | $\sigma$ S3B-CS3S3(-S3B) |
|---|---|---|---|---|---|---|---|---|
| 201812 | -0.002 | 0.073 | -0.004 | 0.072 | 0.001 | 0.072 | -0.002 | 0.072 |
| 201901 | -0.001 | 0.071 | -0.004 | 0.071 | 0.002 | 0.070 | -0.003 | 0.070 |

| 201902 | -0.002 | 0.072 | -0.003 | 0.071 | 0.000 | 0.071 | -0.002 | 0.070 |
| 201903 | -0.003 | 0.074 | -0.005 | 0.075 | 0.000 | 0.072 | -0.004 | 0.073 |
| 201904 | -0.002 | 0.079 | -0.005 | 0.076 | 0.001 | 0.076 | -0.003 | 0.076 |
| all months | -0.002 | 0.074 | -0.004 | 0.073 | 0.001 | 0.072 | -0.003 | 0.072 |

We now notice that CS2S3 freeboards are generally higher than S3B (given by the negative bias for both training and cross-validation comparisons, across all months). The model now appears to fit S3A better than S3B. Rounding does indeed play a role in these statistics, for example, if we increase the number of significant figures for the 'all months' cases $\mu_{CS2-CS2S3}$ and $\mu_{S3A-CS2S3}$ in Table 1, we see that $\mu_{CS2-CS2S3}$ = 0.00078 m and $\mu_{S3A-CS2S3}$ = 0.00024 m. Hence these round to 1 mm and 0 mm respectively (we have included this point in the revised manuscript L254-258). To address the question as to whether the difference in mean between any of the error distributions is significant (e.g., between $\mu_{CS2-CS2S3}$ and $\mu_{S3A-CS2S3}$), we can use a statistical Z-test. This can be computed through the following equation:

$$Z = \frac{\mu_{CS2-CS2S3} - \mu_{S3A-CS2S3}}{\sqrt{\frac{\sigma^2_{CS2-CS2S3}}{n_1} + \frac{\sigma^2_{S3A-CS2S3}}{n_2}}}$$

where $n_1$ and $n_2$ are the number of samples which make up the CS2-CS2S3 and S3A-CS2S3 error distributions respectively. The Z-test allows us to determine whether, based on the available samples from CS2-CS2S3 and S3A-CS2S3, the true means of the two error distributions are likely to be the same (i.e., the true zero-mean Gaussian noise distribution). Note that the Z-test assumes that samples are independent random variables – which is what assume the noise to be. In the example above we find that $Z$ is equal to 2.38, which is equivalent to >99% significance. We therefore do not have evidence to reject the null hypothesis here, and can conclude that the two true means are highly likely to be the same.

- I understand the authors' choice of the cross-validation method, however, I think that both section 4.2 and the conclusions should clearly state that the given estimates of prediction error are based only on validation data from regions below 81.5.N and with a sea ice concentration larger than 75%, since these correspond to areas where the absolute uncertainty is usually the lowest (exception made for the Canadian Archipelago and the Fram Strait, as the authors nicely point out in Section 5). Regions above 81.5.N and with ice concentration between 15% and 75% (including the marginal ice zone) are systematically left out of the cross-validation since:
    - only S3 data are used as a validation
    - according to Lawrence et al. (2019a), diffuse waveforms within grid cells with ice concentration lower than 75% are discarded, which means that no freeboard estimates are available from any of the satellites on a given day where ice concentration falls below 75%.

Thank you for raising this crucial point. We have amended Sect. 4.2 (L275-L278) and the conclusions (L347-L348) to reflect this.

- I would have expected a more significant difference in performance when training the model with CS2 data only, given the lower spatio-temporal coverage when compared with a combined CS2/S3A/S3B training data set. According to your results, a GPR based on CS2 observations alone is able to predict radar freeboard at unobserved locations pretty well (with a 3-4% RMSE increase, from 5.9 to 6.1 cm, when compared to the multi-satellite solution). Do you think this is related to the relatively coarse (50x50 km) grid chosen for your cross-validation? I suggest to add a paragraph in your discussion elaborating on this matter and on the actual advantage of including S3 data in your model training compared with using

only CS2 data. In the light of these results, it would also be interesting to discuss the possibility of using data from the three satellites while reducing the number of days used for model training.

With regards to the benefits of including Sentinel-3 data during the model training, we do see clear improvements in the derived freeboard estimates (see supplementary Fig. S5 of the revised manuscript). In particular, we notice how without S3 data, features such as the 'monkey tail' in the Beaufort Sea are less well defined, and in some cases interpolation artefacts are present (particularly the CS2S3(-S3) case). Furthermore, we also importantly see reduced uncertainty in freeboard by the inclusion of all satellites in the training (see supplementary Fig. S6).

With regards to reducing the number of days for model training, we generated sensitivity tests where we ran interpolations using 3, 5 and 9 days of observations during training (see supplementary Fig. S1). Generally, we see that using only 3 days results in interpolation artefacts in some regions, which are significantly reduced (but not entirely eliminated) by increasing to 5 days. With 9 days of data, we see improved prediction performance and also, on average, reduced prediction uncertainty (see supplementary Fig. S2).

**Assessment of temporal variability**

- This is a nice section highlighting daily variations of regional freeboard estimates and larger discrepancies between CS2S3 predictions and satellite data for sectors like the GIN and the CAA. I suggest to add a couple of statements about the 'Baffin & Hudson' sector. While the average CS2 and S3 freeboard over the entire period agree within 5 mm, they show differences of ~1 cm in December 2018 and March 2019. What do you think might be the reason for this more significant, with respect to other sectors, difference?
  Similar to the regions where we also see differences of ~1cm, e.g. GIN and CAA, we hypothesise that differences in the Baffin & Hudson sector are also a combination of lower latitudes, and therefore sparser sampling and higher uncertainties in interpolated sea level anomalies. We have incorporated this point into Sect. 5 of the revised manuscript.

- I would rephrase line 253 to reflect that the comparison of mean freeboard estimates over the entire observational period performed in this section is mainly a confirmation of your cross-validation results—the average value of a time series alone does not say a lot about temporal variability. I suggest something like: "… Generally, the mean of the CS2S3 time series lies within 3 mm of CS2 and S3, in line with the results of the cross-validation presented in section 4.2. However, ...".
  Thank you for the suggestion, we have amended this statement in the revised manuscript. Please see L297-298.

**Technical corrections**

- L43: according to Lawrence et al. (2019a), the CS2 daily Arctic coverage is lower than 20% up to 82-83.N, not at all latitudes. Also, Tilling et al. (2016) shows Arctic coverage down to a minimum of two days, not one. Please amend this sentence to reflect the content of the cited publications
  Thank you for pointing this out. Please see the revised statement on L43-L45.
- L66/378: the DOI provided for Rasmussen and Williams (2006), a book, points to an article by Matthias Seeger with same title. Please correct the reference
  We have now removed the DOI reference for this text book.

- L104: if you want to be consistent with the platform/sensor notation used for the OSI SAF product, this line should perhaps read: "… from the Nimbus-7/SMMR, DMSP/SSM/I, and DMSP/SSMIS, which are …" → (see https://nsidc.org/data/nsidc-0051 for reference)
  Agreed. We have amended this statement (L113-114).
- L108: you probably mean OSI-403-c? The 403-b product has been superseded and did not include AMSR-2 data
  Yes, thank you for flagging this. We have updated this and changed the reference accordingly (L117).
- L138: "For now…" → "For now, …"
  Agreed. This sounds better. Please see L152.
- L190: "corresponds" → "correspond"
  Thank you, we have now corrected this. Please see L204.
- L291: add comma after "Greenland" → "… and the Greenland, Iceland and Norwegian Seas, …"
  Thank you, we have now corrected this. Please see L355.
- L301: I suggest not use "K" in the final statement → "… and the fact that the covariance structure can take any form, so long as the covariance matrix is symmetric, positive, and semi-definite, means …"
  We agree that this sounds better. We have revised this last sentence (L370-L372) – although we have kept the definition 'positive semi-definite' as this refers to a particular class of matrices whose eigenvalues are strictly non-negative.
- Figure 1: please state which day the sea ice concentration, type (FYI/MYI boundary) and radar freeboard refer to in the example
  Please see the revised Fig. 1 caption, where we have now included the date for which the data correspond to.
- Figure 3: please add the grid resolution (25x25 km) and the day which the radar freeboard estimates and uncertainty correspond to
  Please see the revised Fig. 3 caption, where we have now included the date for which the data correspond to, as well as the grid resolution.
- Figure 6: if the benchmark time series is not explained in the caption, please add a reference to the section 5
  Please see the revised Fig. 7 caption (Fig. 6 in original manuscript), where we have made reference to Sect. 5, with regards to the benchmark time series.
- Figure 7: please write the name of the sectors in full and provide the abbreviations, when used in the text and/or in Figure 6, in parentheses
  Please see the revised Fig. 8 caption (Fig. 7 in original manuscript), where we have provided the full names of each of the sectors, with their abbreviations in parentheses.

  We would like to again thank the reviewer for their invested time in reviewing this work.

**Reviewer Comments 2**

**Summary**

This paper presents a novel approach for estimating daily sea ice freeboard across the Arctic, using the Bayesian inference approach of Gaussian Process Regression (GPR). Benefits of such an approach include 1.) fewer days of data required to estimate Arctic-wide sea ice freeboard and 2.) improved temporal variability of a daily Arctic-wide freeboard product. It's great to see data from CryoSat-2, Sentinel-3A and Sentinel-3B being utilized together in this way. It was also refreshing to read a paper that has a good understanding of the relevant literature and other's methodology, so thank you to the authors for that. I do have a few concerns that should be addressed before publication, and these are summarized in my General and Specific Comments below.

Thank you very much for your enthusiastic feedback, and for taking the time to review our work! Please see our comments below, which we hope address your concerns.

**General Comments**

My one major concern with the paper was the limited assessment of improvements in temporal variability of daily sea ice freeboard (presented in Section 5). The authors show in In Fig. 6 we can see how the day-to-day variability is increased with the CS2S3 product, compared to the CryoSat-2 and Sentinel-3 31-day running means. Then, lines 265-266 state that "A natural question is then whether the variability we see in the time series in Fig. 6 represents real physical signal, or is just noise related to observational uncertainty". I'd suggest that this isn't just a natural question, but really the main question, and one I had throughout the paper until this point. It's really the crux of "why bother doing this work"? While I appreciate the development of innovative methods for improving sea ice products, the reader still needs to be sold on its benefits. Although Figure 6 is very interesting, I'd like to see a more quantitative assessment of the temporal variability from GPR and monthly running means, compared with the benchmark (especially in regions where there are less training data). How much of an improvement in "true" temporal variability does GPR provide? The authors have clearly done most if not all of the relevant work, so please expand. Then, add mention of this in the abstract to strengthen the importance of the study.

Determining whether our interpolation algorithm captures 'real' daily freeboard variability is an important feature of this work, so thank you for helping us to strengthen this section. Although we use the benchmark to ensure that the signal does not originate from spatial sampling, it is difficult to identify what exactly is causing the daily temporal variability. Following your concerns, we have introduced an additional analysis into Section 5, where we compare the evolution of daily radar freeboard anomalies with ERA5 snowfall data (following previous work by Lawrence 2019). With this analysis we see that our estimated daily freeboards are strongly positively correlated with snowfall data, which gives us confidence that the interpolated CS2S3 data able to capture freeboard variability at sub-weekly time scales. Please see L321-L335 of the revised manuscript, as well as the newly introduced Fig. 9. We have also updated the abstract to reflect the results from this analysis.

Reference
Lawrence, I. R.: Multi-satellite synergies for polar ocean altimetry, Ph.D. thesis, UCL (University College London), 2019.

**Specific Comments**

P1 L19: "reductions in the sea ice cover" is too general a statement. Specify what each instrument measures and over what time frame. We haven't seen reduction in thickness from altimetry for four decades, or in summer. We also haven't seen reduction in extent from altimetry. So, please be more specific here to avoid confusion.

Agreed. We have revised this section. Please see L20-L22 of the revised manuscript.

P2 L23: Are AGU talks suitable references (I'm not sure on TC's stance on this)? If so, please provide a link to the publicly available version of the talk.

We have removed this reference from the article in the revised manuscript (L23).

P2 L31: I'd consider Allard et al. (2018) to be a key paper that's missing here

Thank you for pointing us to this paper, we have added this reference to L31 of the revised manuscript.

P2 L35: Snow depth is also assumed

Agreed. We have added this assumption to our list of points on L36 of the revised manuscript

P2 L43: I believe Tilling et al. (2016) was 2 days

Thank you for this. See the revised comment on L44-L45 of the revised manuscript.

P2 L55: Change "containing" to "assimilating", for clarity

Agreed, assimilating is clearer. We have changed this on L55.

P3 L60: This first sentence doesn't add anything. I suggest using at this point to highlight the benefits of the GPR method vs. a monthly moving average that is very simple the produce. It's not just the need for a daily freeboard product based on observations that is well motivated, but a daily freeboard product that more accurately represents temporal variability. This new approach can (in theory) provide both.

Agreed. Please see our revisions of this section on L64-L66.

P3 L75: For me, Section 5 is currently insufficient at providing "an assessment of the improved temporal variability achieved by the use of a daily product". See my General Comments above.

We hope with our revisions to Section 5, that you are happy for us to leave this statement as per the original manuscript.

P4 L99: Introduce the "CS2S3" acronym here

We have added the acronym to the revised manuscript in L102.

P4 L120: On average, what percentage are co-located?

As RC1 also raised a question related to co-located observations, we have incorporated a few sentences describing how they are handled in our workflow and what percentage of observations are co-located on average. Please see L138-L141 of the revised manuscript.

**References**

Allard, R. A., Farrell, S. L., Hebert, D. A., Johnston, W. F., Li, L., Kurtz, N. T., et al. (2018). Utilizing CryoSat-2 sea ice thickness to initialize a coupled ice-ocean modeling system. Advances in Space Research, 62(6), 1265–1280.

Tilling, R. L., Ridout, A., & Shepherd, A. (2016). Near Real Time Arctic sea ice thickness and volume from CryoSat-2. The Cryosphere, 10, 2003–2016.

We would like to again thank the reviewer for their invested time in reviewing this work.